# On the influence of vertical mixing, boundary layer schemes, and temporal emission profiles on tropospheric $NO_2$ in WRF-Chem - Comparisons to in-situ, satellite, and MAX-DOAS observations

Leon Kuhn[1, 2], Steffen Beirle[2], Vinod Kumar[2], Sergey Osipov[4, 2], Andrea Pozzer[2, 6], Tim Bösch[5], Rajesh Kumar[3], and Thomas Wagner[1, 2]

[1]Institute for Environmental Physics, University of Heidelberg, Germany
[2]Max Planck Institute for Chemistry, Mainz, Germany
[3]National Center for Atmospheric Research, Boulder, United States of America
[4]King Abdullah University of Science and Technology, Thuwal, Saudi Arabia
[5]Institute for Environmental Physics, University of Bremen, Germany
[6]Climate and Atmosphere Research Center, The Cyprus Institute, Nicosia, Cyprus

**Correspondence:** Leon Kuhn (l.kuhn@mpic.de)

**Abstract.** We present WRF-Chem simulations over central Europe with a spatial resolution of $3 \, \text{km} \times 3 \, \text{km}$ and focus on nitrogen dioxide ($NO_2$). A regional emission inventory, issued by the German Environmental Agency, with a spatial resolution of $1 \, \text{km} \times 1 \, \text{km}$ is used as input. We demonstrate by comparison of five different model setups, that significant improvements in model accuracy can be achieved by choosing the appropriate boundary layer scheme, increasing vertical mixing strength, and/or

tuning the temporal modulation of the emission data ("temporal profiles") driving the model. The model setup with improved vertical mixing is shown to produce the best results. Simulated $NO_2$ surface concentrations are compared to measurements from a total of 275 in-situ measurement stations in Germany, where the model was able to reproduce average noontime $NO_2$ concentrations with a bias of ca. $-3 \, \%$ and $R = 0.74$. The best agreement is achieved when correcting for the presumed $NO_y$ cross sensitivity of the molybdenum-based in-situ measurements, by computing an $NO_y$ correction factor from modelled PAN

and $HNO_3$ mixing ratios. A comparison between modelled $NO_2$ vertical column densities (VCDs) and satellite observations from TROPOMI (TROPOspheric Monitoring Instrument) is conducted, with averaging kernels taken into account. Simulations and satellite observations are shown to agree with a bias of $+5.5 \, \%$ and $R = 0.87$ for monthly means. Lastly, simulated $NO_2$ concentration profiles are compared to noontime $NO_2$ profiles obtained from Multiaxis Differential Optical Absorption Spectroscopy (MAX-DOAS) measurements at five locations in Europe. For stations within Germany, average biases of $-25.3 \, \%$ to

$+12.0 \, \%$ were obtained. Outside of Germany, where lower resolution emission data was used, biases of up to $+50.7 \, \%$ were observed. Overall, the study demonstrates the high sensitivity of modelled $NO_2$ to the mixing processes in the boundary layer and the diurnal distribution of emissions.

# 1 Introduction

Modelling of regional atmospheric chemistry and transport (RCT) is an important discipline in the field of air quality research. While observational data is often only available at coarse spatial or temporal resolutions, model data can be generated on regular grids of much higher spatio-temporal resolution. Modern RCT models can therefore be used to systematically investigate the processes of transport and (photo)-chemical conversion that trace gases are subject to upon emission into the atmosphere. Most importantly, however, they allow for predictions of trace gas concentrations when observational data is not available and operational air quality forecasting. Thus, they give valuable insight into the dynamics of air quality in polluted regions of the earth. Examples for state of the art RCT models are WRF-Chem (Grell et al., 2005), COSMO/MESSy (Kerkweg and Jöckel, 2012), Lotos-Euros (Manders et al., 2017), CAM-chem (Emmons et al., 2020), and CHIMERE (Menut et al., 2021).

Nitrogen dioxide ($NO_2$) is one of the most relevant chemical species for air quality in polluted regions. It is toxic to humans and acts as a precursor for ozone ($O_3$), a key pollutant of urban smog. The hazardous impact of $NO_2$ on human health has been widely recognized among the scientific community (see e.g. Faustini et al. (2014); Mills et al. (2015); Chowdhury et al. (2021)). Monitoring and predicting realistic distributions of $NO_2$ in the troposphere is therefore of ongoing political and scientific interest.

Past modelling efforts with focus on tropospheric $NO_2$ have typically resulted in very similar general outcomes: while researchers found good agreement between modelled and observed meteorological data (such as wind speeds and air temperature), systematic disagreements between modelled surface $NO_2$ concentrations and in-situ observations were found. Visser et al. (2019) report on the results of a WRF-Chem simulation using the Carbon Bond Mechanism Z (CBM-Z) over central Europe in which noontime $NO_2$ surface concentrations and vertical column densities (VCDs) were underestimated by 38.5 % and approximately 15 %, respectively. In a comparison of monthly mean $NO_2$ VCDs, $R$-values between 0.82 and 0.92 were obtained. The authors identify an underestimation of soil emissions in their emission inventory (TNO-MACC-III, short for "Monitoring Atmospheric Composition and Climate" by the Netherlands Organisation for Applied Scientific Research) as a possible explanation. Kuik et al. (2016) present a WRF-Chem simulation with the Regional Acid Deposition Model 2 (RADM2) and the same emission inventory as Visser et al. (2019) over the region of Berlin, Germany, and observe even stronger underestimations of surface $NO_2$ by more than 50 % during daytime and a strong overestimation at night-time in a similar comparison. The study reveals that increasing the spatial resolution (including downscaling of emission data) of the model from $15\,\mathrm{km} \times 15\,\mathrm{km}$ to $1\,\mathrm{km} \times 1\,\mathrm{km}$ slightly improves agreement, but not to a satisfying degree. In a subsequent publication (Kuik et al., 2018), the authors attribute the disagreements to underestimations in the emission data. Poraicu et al. (2023) show a WRF-Chem simulation with the CBM-Z mechanism over a domain in Belgium, where simulated $NO_2$ showed daytime underestimations of up to 25.1 % and nighttime overestimations of up to 77.3 %. The authors experiment with the choice of the boundary layer scheme, as well as a correction factor for cross sensitivities of the reference in-situ measurements to other nitrogen compounds. Although slight improvements were achieved, the described model bias persists. Du et al. (2020) demonstrate that tuning the vertical mixing parametrization of different boundary layer schemes in WRF-Chem drastically reduces model biases (shown for PM2.5 over China), particularly at nighttime during summer months. Mar et al. (2016) study the influence

of the chemical mechanism on modelled $O_3$ and $NO_2$ by direct comparison of the mechanisms MOZART (Model for OZone and Related chemical Tracers) and RADM2. While the two mechanisms were found to produce significantly different results for $O_3$, the differences in modelled $NO_2$ were much smaller. On average, the $NO_2$ concentrations obtained from MOZART were 2 $\mu g\,m^{-3}$ larger than those obtained from RADM2. However, a study based on box-model simulations by Knote et al. (2015) reveals much larger discrepancies between chemical mechanisms of up to 25 % for $NO_x$ and 100 % for $NO_3$, which plays a significant role in nighttime $NO_x$ chemistry. Furthermore, some chemical mechanisms were found to be outdated, e.g. with respect to organic nitrate chemistry.

Kumar et al. (2021) demonstrate in a simulation with the MECO(n) model system over Germany using a mixture of TNO-MACC-III and regional emission data, that agreement between modelled $NO_2$ concentrations and in-situ observations improves greatly when diurnal and seasonal variability is added to the yearly resolved emission data using hourly and monthly weighting factors ("temporal profiles"). This temporal upsampling has become common practice among the air quality modelling community and standard values for such temporal profiles have been established (see Crippa et al. (2020b); Kumar et al. (2021) and the references within). Based on their efforts to reduce model bias by other means, Poraicu et al. (2023) conclude that the mismatch between modelled and observed diurnal cycle of surface $NO_2$ could relate to faulty diurnal emission profiles and/or to insufficient vertical mixing during the night.

A number of publications show that the observed daytime low bias of modelled surface $NO_2$ could relate to systematic flaws in the ground based in-situ measurements used as reference. Conventional in-situ methods often utilize molybdenum converters, which were found to be cross sensitive to other reactive nitrogen species, such as PAN, $HNO_3$, and alkyl nitrates, summarized as $NO_y$. This issue was discussed e.g. by Dunlea et al. (2007), Steinbacher et al. (2007), Lamsal et al. (2008), Boersma et al. (2009), and Villena et al. (2012), who found biases reaching up to a factor of 4 with a strong correlation to $O_3$ (which again correlates with photochemical activity). Poraicu et al. (2023) attempt to account for such cross sensitivities by computing a correction factor based on simulated surface mixing ratios of PAN and $HNO_3$. The method contributes to resolving the daytime low bias of the model, but is not helpful with respect to the even larger high bias at nighttime. In Europe, in-situ measurements of $NO_2$ must conform to regulations defined by the European Norms 14221, 14181, and 15267, which require empirical evidence that the instrument in question is unbiased against direct (e.g. spectroscopic) measurements of $NO_2$. Such conformity assessments are conducted and thoroughly protocolled by technical inspection associations (such as the TÜV for the in-situ measurements in Germany, which are used in this article, see German Environmental Agency (a)). There is a clear conflict between the overestimations reported in the scientific literature and the proclaimed conformance to the European regulations and the true magnitude of the problem remains up for question.

Altogether the contemporary literature comes to a clear consensus: Surface $NO_2$ concentrations in RCT simulations are typically underestimated at daytime and overestimated at nighttime. The phenomenon was observed consistently across different combinations of geographical domains, model resolution, emission inventories, boundary layer schemes, chemical mechanisms, and reference data sources used in the past (see Visser et al. (2019); Kuik et al. (2016); Kuik et al. (2018); Poraicu et al. (2023)). Comparisons to other observational datasets (mainly satellite observations) do occur in literature, and show

generally better agreement. However, satellite measurements often yield only a single measurement of the vertically integrated concentration per day, i.e. they do not cover the diurnal cycle.

In this paper, we show the results of a WRF-Chem simulation over central Europe for the month of May 2019 with a spatial resolution of $3\,\text{km} \times 3\,\text{km}$. The goal is to quantify the level of agreement between simulated $NO_2$ concentrations and VCDs and the corresponding results from different observational datasets. For this purpose our simulation results are compared to three reference datasets:

1. Surface $NO_2$ concentrations measured by a network of in-situ instruments over Germany, operated by the German Environmental Agency (UBA, see German Environmental Agency (b))

2. Tropospheric $NO_2$ VCDs measured by TROPOMI (TROPOspheric Monitoring Instrument) on the Sentinel 5 precursor satellite, specifically the processor version 20400. This includes recomputing the air mass factors (AMFs) of the retrieval based on our simulation results.

3. $NO_2$ concentration profiles obtained from five MAX-DOAS instruments that partake in the FRM4DOAS project (see Fayt et al., 2021)

In the scope of a sensitivity analysis, we test whether the model bias observed in the diurnal cycle of surface $NO_2$ can be resolved by

1. correcting for the $NO_y$ cross sensitivity of the UBA in-situ measurements, based on modelled PAN and $HNO_3$

2. comparing different boundary- and surface layer schemes

3. implementing tuned mixing, as proposed by Du et al. (2020)

4. tuning the temporal emission profiles of the most dominant emission sectors

or a combination of the above.

The article is structured as follows: Section 2 describes the different setups of our WRF-Chem simulation and the pre-processing of emission data in detail. Section 3 compares the individual model runs to each other and the above-mentioned observational datasets. Possible explanations for the observed differences are given from a technical perspective. Section 4 presents a conclusion and discussion of the results.

## 2 WRF-Chem simulation setup

WRF-Chem (Weather Research and Forecasting model with Chemistry, see Grell et al., 2005) is a mesoscale RCT model. We run WRF-Chem version 4.2.2 on a twofold nested domain over central Europe for the month of May, 2019, see Fig. 1. The spatial resolutions of the outer and inner domain (called D1, and D2 from hereon) are $15\,\text{km} \times 15\,\text{km}$, and $3\,\text{km} \times 3\,\text{km}$ (with $320 \times 245$ pixels and $500 \times 430$ pixels, respectively). The temporal resolution of the simulation is 60 seconds on D1 and

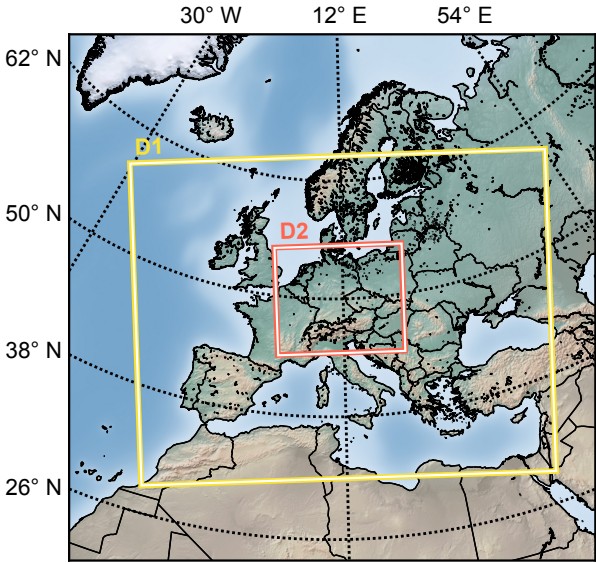

**Figure 1.** Geographical coverage of the WRF-Chem simulation. The spatial resolutions of the outer domain D1 and the inner domain D2 are $15 \text{ km} \times 15 \text{ km}$, and $3 \text{ km} \times 3 \text{ km}$, respectively.

12 seconds on D2. Output files are written daily for D1 and hourly for D2. The simulation uses the Thompson microphysics scheme (see Thompson et al., 2008), the RRTMG (Rapid Radiative Transfer Model for General Circulation Models long- and shortwave radiation scheme, see Iacono et al., 2008), the Monin-Obukhov similarity scheme for surface layer modelling (see Monin and Obukhov, 1954), the NOAH Land-Surface Model (see Niu et al., 2011), the YSU boundary layer scheme (see Hong, 2010), and the Grell-Dévényi ensemble scheme for cumulus modelling (see Grell and Dévényi, 2002). For modelling of chemistry, the MOZART-4 chemical mechanism (see Emmons et al., 2010) is coupled to the GOCART aerosol mechanism (see Chin et al., 2000) along with the TUV full photolysis scheme (Madronich (1987); Tie et al. (2003)), which deploys climatological $O_3$ and $O_2$ columns. Dry deposition is calculated according to Wesely (1989). Spectral nudging (see e.g. Omrani et al., 2012) to ERA5 reanalysis data (see Hersbach and Dee, 2017) is used on D1. The simulation uses a total of 43 vertical levels in terrain-following coordinates (see Table A1 for the layer heights up to 6 km). Both domains D1 and D2 receive initial conditions from the CAM-chem model (Emmons et al., 2020). Additionally, CAM-chem yields the boundary conditions for D1. D2 receives boundary conditions online from the WRF-Chem simulation running on D1. We refer to this base model setup as "S-YSU".

## 2.1 Emission preprocessing

The WRF-Chem simulation is driven by different emission inventories for different emission sectors. Emissions from biomass burning are taken from the Fire Inventory from NCAR (FINN, see Wiedinmyer et al., 2011) with a spatio-temporal resolution of $1 \text{ km} \times 1 \text{ km}$ and 24 hours. Biomass burning emissions are assumed to have a diurnal variation with a peak in the early

afternoon (1 pm local time) and are distributed vertically in the model following the plumerise parametrization of Freitas et al. (2007). Biogenic emissions are computed online using an implementation of the MEGAN model (see Guenther et al., 2006). For anthropogenic emissions a combination of two emission inventories is used: Over Germany, an inventory of high spatial resolution (1 km × 1 km, resampled to $0.01° × 0.01°$ or $\sim 1.1$ km × 0.7 km) is provided by the UBA (see Strogies et al., 2020). From hereon, this emission inventory will be referenced as UBA-E. Using the UBA-E emission data over Germany enables modelling of $NO_2$ distributions at a high spatial resolution. Outside of Germany the EDGARv5 emission inventory with a moderate spatial resolution ($0.1° × 0.1°$ or $\sim 11$ km × 7 km) is used (see Crippa et al., 2020a). Since UBA-E does not include organic anthropogenic emissions, EDGARv5 is used on the entirety of D1 and D2 for organic species. Non-methane volatile organic compounds (NMVOCs) are provided as lumped species and are speciated according to Huang et al. (2017). An alternative to EDGARv5 would have been the TNO-MAC-III emission inventory (Kuenen et al., 2014) which comes at a higher spatial resolution of $0.0625° × 0.125°$ but was only available for the year 2011 at the time the simulation was run. Since EDGARv5 was available for 2015 and $NO_x$ emissions have steadily decreased over the past years (see e.g. Anenberg et al., 2022), EDGARv5 was considered a more reasonable choice.

EDGARv5 (for 2015) and UBA-E have time resolutions of 1 month and 1 year, respectively. However, many emissions follow strong diurnal and seasonal patterns. For example, emissions from car traffic are expected to be lower at nighttime and higher at daytime and agricultural emissions typically occur in specific months of the year. EDGARv5 can still resolve emission variations on a monthly time-resolution but is incapable of resolving diurnal patterns. UBA-E provides only annual emissions, therefore without any temporal pattern. The solution to this problem is to scale the coarsely resolved emission data to an hourly resolution using presumed hourly, daily, and monthly emission scaling factors ("temporal profiles"). The emission rate $E_{X,k}(m, d, h, \mathrm{lat}, \mathrm{lon})$ of a species $X$ from sector $k$ at month $m$, day $d$, and hour $h$ at fixed latitude lat and longitude lon is given as

$$E_{X,k}(m, d, h, \mathrm{lat}, \mathrm{lon}) = \hat{E}_{X,k}(\mathrm{lat}, \mathrm{lon}) \cdot p_{\mathrm{monthly},k}(m) \cdot p_{\mathrm{daily},k}(d) \cdot p_{\mathrm{hourly},k}(h) \tag{1}$$

where $\hat{E}_{X,k}$ denotes the total emissions of species $X$ from sector $k$ in the emission inventory and $p_{\mathrm{monthly}}$, $p_{\mathrm{daily}}$, and $p_{\mathrm{hourly}}$ the monthly, daily, and hourly temporal profiles. The individual profiles are normalized to 12 (annual cycle), 7 (weekly cycle), and 24 (diurnal cycle), respectively. Because different emission sectors follow vastly different temporal patterns, the temporal profiles are defined for each sector individually. An overview of the emission sectors is given in Table 1. The total emission rate of species $X$ is obtained by summation over all emission sectors $k$

$$E_X(m, d, h, \mathrm{lat}, \mathrm{lon}) = \sum_k \hat{E}_{X,k}(\mathrm{lat}, \mathrm{lon}) \cdot p_{\mathrm{monthly},k}(m) \cdot p_{\mathrm{daily},k}(d) \cdot p_{\mathrm{hourly},k}(h) \tag{2}$$

Some species are only implicitly contained in the emission inventories. For example, EDGARv5 and UBA-E specify $NO_x$ emissions, but the partitioning into $NO$ and $NO_2$ must be chosen by the user via a speciation profile $p_{\mathrm{spec}}$. Equation (2) then generalizes to

$$E_X(m, d, h, \mathrm{lat}, \mathrm{lon}) = \sum_k \hat{E}_{X_{\mathrm{lump}},k}(\mathrm{lat}, \mathrm{lon}) \cdot p_{\mathrm{monthly},k}(m) \cdot p_{\mathrm{daily},k}(d) \cdot p_{\mathrm{hourly},k}(h) \cdot p_{\mathrm{spec},k}(X) \tag{3}$$

**Table 1.** Emission sectors in the UBA-E and EDGARv5 emission inventories

| Name | Contribution[†] (UBA-E) [%] | Contribution[†] (EDGARv5) [%] |
|---|---|---|
| traffic (no resuspension) | 43.8 | 38.7 |
| power industry | 18.5 | 15.0 |
| agricultural soils | 10.1 | 4.5 |
| energy for buildings | 7.3 | 6.6 |
| manufacturing industry | 7.2 | 15.8 |
| non-metallic minerals production | 2.5 | 0.0 |
| production of chemicals | 2.4 | 0.3 |
| shipping | 2.1 | 2.1 |
| iron and steel production | 1.8 | < 0.1 |
| oil refineries and transformation industry | 1.7 | 1.8 |
| aviation landing and take-off | 1.2 | 1.8 |
| railways, pipelines, and off-road transport | 0.9 | 2.3 |
| production of food, pulp, and paper | 0.3 | 0.3 |
| manure management | 0.1 | 1.1 |
| fuel exploitation | 0.1 | 0.0 |
| solid waste incineration | 0.1 | 0.1 |
| non-ferrous metal production | 0 | < 0.1 |
| non-energy use of fuels | < 0.1 | 0.0 |
| agricultural waste burning | N/A | 0.3 |
| fossil fuel fires | N/A | < 0.1 |
| aviation climbing and descent | N/A | 5.9 |
| aviation cruise | N/A | 3.4 |

N/A = "not available"

[†] Relative contribution of this sector to the overall emissions (yearly for UBA-E, and for the month of May for EDGARv5) of $NO_x$

where $X_{lump}$ is the lump of species that contains species $X$ (i.e. $X_{lump} = NO_x$ when $X = NO_2$).

In principle, WRF-Chem also supports vertical distribution of trace gas emissions. This is reasonable to consider, given that many strong emissions, like those from combustion stacks, take place at elevated altitudes. In analogy to the mentioned temporal and speciation profiles, this can be modelled using an additional vertical emission profile. Suggestions for vertical emission profiles are given in Bieser et al. (2011) and Pozzer et al. (2009). The optimization of vertical emission profiles is not in the scope of our study, because the majority of the observational data used (in-situ measurements and most satellite

observations) stems from background locations, where almost all emissions occur at the surface and transport and vertical

mixing further minimize the influence of height-distributed emissions. Hence, all emissions are injected into the lowest model layer (0 - 8 m).

## 2.2 Model variants with different boundary layer and surface layer schemes

We conduct two additional model runs with different combinations of planetary boundary layer (PBL) and surface layer scheme:

1. The Mellor-Yamada-Janjić scheme (MYJ, see Mesinger, 2020; Janjić, 1994), coupled with the Monin-Obukhov (Janjić Eta) similarity scheme. We refer to this model setup as "S-MYJ".

2. The Bougeault-Lacarrere scheme (BouLac, see Bougeault and Lacarrere, 1989), coupled with the Monin-Obukhov scheme with Carlson-Boland viscous sub-layer. We refer to this model setup as "S-BouLac".

This choice of schemes was motivated by the the WRF-Chem user guide (which recommends the YSU and MYJ boundary layer schemes, see University Corporation for Atmospheric Research), as well as the findings of Poraicu et al. (2023), who demonstrate that the BouLac boundary layer scheme tends to produce particularly low $NO_2$ concentrations at nighttime.

## 2.3 Model variant with tuned vertical mixing

The work of Du et al. (2020) demonstrates, that the diurnal cycle of PM2.5 can be reproduced much more accurately by tuning the vertical mixing within WRF-Chem. Given that PM2.5 and trace gases are subject to the same mixing routine in WRF-Chem, it is highly plausible that their approach has a similar effect on the diurnal cycle of $NO_2$.

Mixing in WRF-Chem is computed in two steps: First, a mixing coefficient $k_h$ (called "EXCH_H" in the WRF-Chem registry) is computed by the PBL scheme. Then, a mixing routine is called, which dilutes the trace gas concentration of each model layer into its neighbours based on the magnitude of $k_h$. The mixing routine implements a crucial manipulation of $k_h$: Depending on a coarse classification of the model cell as either "rural" or "urban", $k_h$ is clipped to a minimum of $1\,\mathrm{m^2\,s^{-1}}$ or $2\,\mathrm{m^2\,s^{-1}}$ for the lowest model layers. This is fundamental with regards to modelling surface concentrations, seeing that the boundary layer schemes shown in Du et al. (2020) (and those shown within this paper) tend to produce mixing coefficients smaller than $1\,\mathrm{m^2\,s^{-1}}$ at the surface, particularly at nighttime. In essence, this implementation results in increased vertical mixing, and the effect is strongest during the night. The WRF-Chem source code encourages the user to tune this enhancement further (see the WRF-Chem source code file "chem/dry_dep_driver.F", where the described mixing procedure is implemented). However, the parametrization is hard-coded, and can only be changed by complete recompilation of WRF-Chem. Following the recommendation of Du et al. (2020), we present a model run where the clipping of $k_h$ is enhanced to $5\,\mathrm{m^2\,s^{-1}}$ everywhere. We refer to this model setup as "S-YSU+5".

## 2.4 Model variant with tuned diurnal emission profiles

Another approach to reducing the bias of the modelled diurnal $NO_2$ cycle is to tune the hourly profiles used in the preprocessing of the emission data. The goal is to find temporal profiles that minimize the model's mean $NO_x$ bias over the course of the day.

The mean relative bias is computed as

$$\text{bias} = \frac{1}{N} \sum_{i=1}^{N} \frac{x_{\text{sim},i} - x_{\text{obs},i}}{x_{\text{obs},i}} \tag{4}$$

where $x_{\text{obs},i}$ denotes the $i$-th observation and $x_{\text{sim},i}$ the corresponding simulated value. We use the observations from in-situ measurements of background $NO_x$ surface concentrations as reference. A detailed explanation of the dataset is given in sect. 3.1. Equation (3) hints towards the complexity of the optimization problem: Because the emission inventories include dozens of emission sectors, each with their own emission profiles, this poses an optimization problem with many degrees of freedom. In addition, a single WRF-Chem simulation of just one month takes days to finish even on modern supercomputers. This circumstance makes it nearly impossible to optimize the emission profiles using standard methods like gradient descent. For our simulation we have therefore optimized the emission profiles empirically. By "empirical optimization" we mean the iterative process of running WRF-Chem, evaluating the simulation results against in-situ observations, and slightly nudging the temporal profiles in a direction in which better agreement between simulation and observations can be expected. Due to the short lifetime of $NO_x$ the observed concentrations follow the temporal profiles without much delay. As an initial starting point, the temporal profiles from Kumar et al. (2021) were used. Although the proposed optimization method is rather unconventional, it has a few important benefits. Consider the following:

1. The hourly profiles of many sectors have characteristic shapes, e.g. a peak in the hourly profile of the traffic sector during the morning rush hour. These should be at least coarsely preserved during the optimization process in order to maintain realistic emission behaviour.

2. Because the optimization problem is ill-posed, it is often unclear, of which sector the profiles should be tuned further in order to improve the simulation. Sometimes, the spatial distribution of a specific emission sector matches that of the model error, indicating that this sector should be prioritized.

3. Gradient-based optimization methods depend on hyperparameters, such as the step size. If these are not picked correctly from the start, the optimization may converge slowly or diverge entirely. This becomes infeasible, when a single forward run takes days to compute.

Using a conventional optimization routine, where the gradient of a loss function determines the outcome of a single optimization step, would require to encode aspects 1-3 in the form of mathematical constraints. This makes a rigorous treatment of the problem extremely complex. The empirical optimization approach, however, does not require a mathematical formulation and can thus take the discussed aspects into consideration more easily. We run the optimization under the following conditions:

1. Only the most relevant emission sectors are modified during the process. According to Table 1 these are: traffic (no resuspension), power industry, agricultural soils, energy for buildings, and manufacturing industry.

2. Speciation of lumped species follows the recommendations of Huang et al. (2017) and is not further optimized. For $NO_x$, the partitioning is assumed to be 87.5 % NO and 12.5 % $NO_2$. This choice reflects the fact that $NO_x$ from combustion

**Table 2.** Overview of the different model setups used within our study

| Model setup | Difference to standard setup | Boundary layer namelist option | Surface layer namelist option |
|---|---|---|---|
| S-YSU | — | 1 | 1 |
| S-MYJ | using Mellor-Yamada-Janjić scheme | 2 | 2 |
| S-BouLac | using Bougeault-Lacarrere scheme | 8 | 1 |
| S-YSU+5 | clipping of mixing coeff. $k_h$ set to 5 m$^2$ s$^{-1}$ | 1 | 1 |
| S-YSU-TP | using tuned hourly emission profiles | 1 | 1 |

processes is mostly emitted as NO, which oxidizes to $NO_2$ over time. Literature values for typical $NO_2/NO_x$ ratios in anthropogenic emissions range from lower values (e.g. 5.3 %, as reported by Wild et al. (2017) and 7.8 %, as reported by Jimenez et al. (2000)) to much higher values (e.g. 39 %, as reported by Richmond-Bryant et al. (2017) and 36 %, as reported by Costantini et al. (2016)).

3. In order to improve generalization of the temporal profiles, the optimization is performed using data from May 2018. Additionally, we accelerate the optimization by only using two weeks of simulation time for each optimization step. The final temporal profiles are evaluated in a full-month simulation for the year 2019, as described in sect. 2.

4. The $NO_y$ cross sensitivities of the reference measurements are not taken into account during the optimization process. Their influence is described in more detail in sect. 3.1.

Figure 2 gives an overview of the optimization process in a total of 3 iterations. The monthly and weekly emission cycle was taken directly from Kumar et al. (2021). Then, the hourly profiles were adjusted as to compensate the model bias. For example, in step 1 (red line), only a single change was made to the traffic sector profile (subfigure 2a) in order to boost simulated $NO_x$ values in the morning ($\sim$ 6 AM). The principle was extended to the remaining hourly profiles in the next two steps. We refer to this model setup as "S-YSU-TP". Table 2 gives an overview of the five different model setups used within our study.

## 3 Results

### 3.1 Comparison of surface concentrations

In-situ measurements of trace gas surface concentrations in Germany are conducted by the UBA. They are available for $NO_2$, NO and $O_3$ as hourly mean values. A total of 434 UBA measuring stations are distributed over Germany. 92 % of the stations use a chemiluminescence based measuring method. The remaining 8 % use other methods (cavity enhanced phase shift spectroscopy, diffuse sampling, or photolysis conversion). Of all UBA stations, 63 % are classified as "background", 30 % as "traffic", and 7 % as "industry". In this study we only take "background" stations into consideration. This choice was made seeing that $NO_2$ concentration can vary strongly near traffic over distances of only 10 - 100 m (see e.g. Beckwith et al., 2019) and

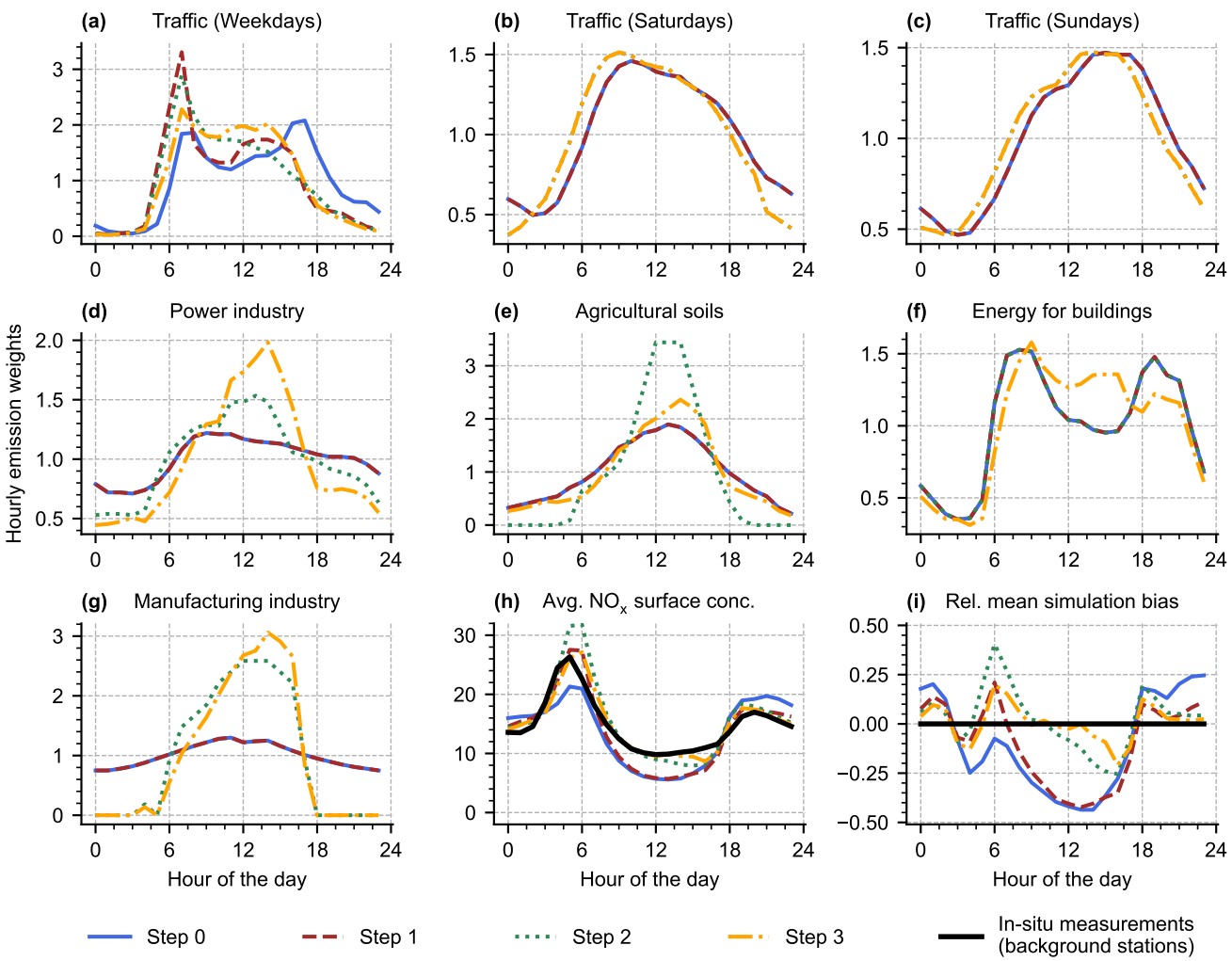

**Figure 2.** Overview of the optimization process for the diurnal emission profiles based on the in-situ observations of May 2018. The profiles used by Kumar et al. (2021) are drawn in blue and the optimized profiles used in S-YSU-TP are drawn in orange.

timescales of a few minutes. Given our simulation's spatial resolution of $3\,\mathrm{km} \times 3\,\mathrm{km}$, it is expected to show poor agreement with the traffic stations.

In order to compare the simulated surface concentrations to the in-situ measurements, they are interpolated from the WRF-Chem simulation grid to the geolocations of the UBA stations. As introduced in sect. 1, the molybdenum-based chemiluminescence method for $NO_2$ detection is likely cross sensitive to $NO_y$. We account for this by computing a correction factor

$$F = 1 + \frac{0.95 \cdot [\mathrm{PAN}] + 0.35 \cdot [\mathrm{HNO_3}]}{[\mathrm{NO_2}]} \tag{5}$$

as proposed by Lamsal et al. (2008) and Poraicu et al. (2023). Here, [PAN], [HNO$_3$], and [NO$_2$] denote the volume mixing ratios of PAN, HNO$_3$, and NO$_2$, respectively, and are taken from the WRF-Chem simulation output. According to Lamsal et al. (2008), a more precise formulation of eq. (5) would require additional consideration of all alkyl nitrates, but those are not available in the MOZART-4 chemical mechanism, and $F$ is expected to be dominated by HNO$_3$ (see Dunlea et al., 2007). Elshorbany et al. (2012) showed, that the contribution of HONO to $F$ can be expected to be in the range of 2-6 %. The correction factor is applied by multiplying it with the simulated $NO_2$ concentrations, if the collocated measuring station uses a molybdenum converter.

Figure 3 shows the average surface concentrations of $NO_2$, $NO$, $NO_x$, and $O_3$ of the base simulation run S-YSU. The left panel (subfigures 3a-d) is restricted to the first ten days of the simulation (01 May 2019 - 10 May 2019) for easier readability. The right panel (subfigures 3e-h) shows the average diurnal concentrations obtained from averaging over all days of the simulation. The model is evaluated over three time spans: noontime (12 PM), daytime (6 AM - 8 PM), and nighttime (9 PM - 5 AM). The diurnal cycle of simulated $NO_2$ depicts a moderate low bias of -15.7 % (1.3 $\mu$g m$^{-3}$, $R = 0.75$) at noontime, an overall daytime bias of +18.4 % (1.9 $\mu$g m$^{-3}$, $R = 0.80$) and a strong positive bias at nighttime of +53.1 % (7.6 $\mu$g m$^{-3}$, $R = 0.49$). Application of the $NO_y$ correction factor helps to alleviate the noontime low bias to +3.1 % (0.2 $\mu$g m$^{-3}$, $R = 0.75$), but increases the daytime and nighttime biases to +30.9 % and +60.5 %, respectively. The diurnal cycle of NO is reproduced with deviations of similar magnitude: From midnight to 6 AM, when the characteristic morning peak builds up, the simulation results show a low bias of -40.3 % (1.5 $\mu$g m$^{-3}$, $R = 0.62$). For the remaining hours of the day, NO is reproduced with a low bias of -16.3 % (0.3 $\mu$g m$^{-3}$, $R = 0.83$). The diurnal cycle of $NO_x$ is dominated by $NO_2$, except for the brief morning period around 5 AM, when the NO concentration peaks. The $NO_x$ bias of the model results in -4.2 % at noontime, +22.2 % during the day, and +43.7 % during the night, which is similar to the model's $NO_2$ bias. $O_3$ is generally overestimated by the simulation, by a mostly constant difference of approximately 13 $\mu$g m$^{-3}$ ($R = 0.78$). A version of Fig. 3 showing traffic stations instead of background stations can be found in appendix A (Fig. A1). The diurnal $NO_2$ cycle measured by the traffic observations has a fundamentally different shape compared to the background observations, with practically no noontime low and no evening peak. As expected, the modelled and observed concentration cycles do not agree well for the traffic stations.

The $NO_y$ correction factor $F$ has a significant impact on the noontime agreement to the observations. Figure 4 depicts the average diurnal cycle of $F$ with a clear correlation to the $O_3$ surface concentration. This is expected, seeing that both $O_3$,

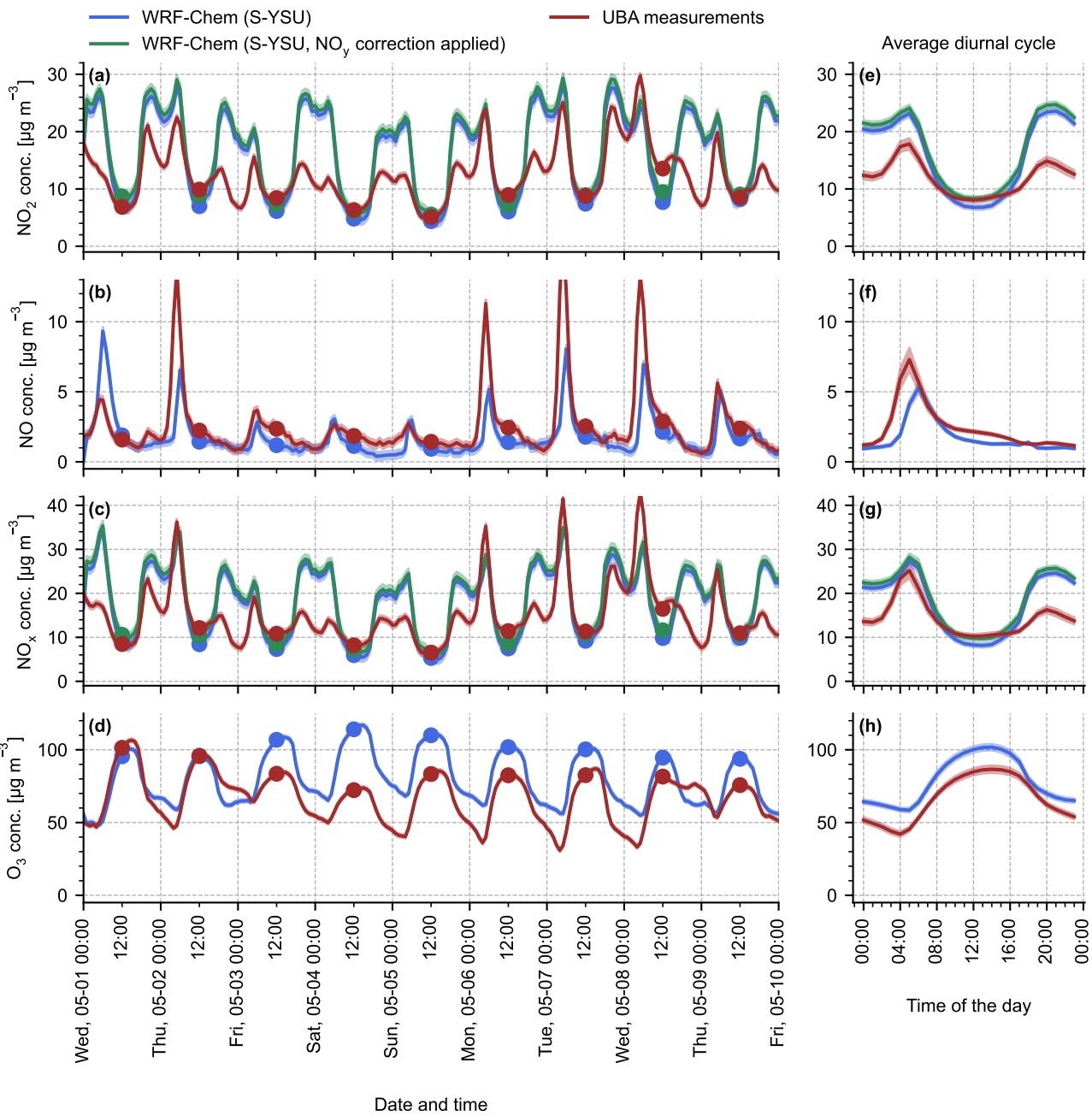

**Figure 3.** Time series of $NO_2$, $NO$, $NO_x$, and $O_3$ surface concentrations. The red lines show reference values obtained from the UBA background in-situ stations. The blue lines show the simulation results of simulation run S-YSU. **(a)** - **(d)** display a time series spanning multiple days (01 May 2019 - 10 May 2019) and **(e)** - **(h)** display the corresponding average concentration values over the entire month of May 2019. **(a)** and **(e)** show the diurnal cycle of simulated $NO_2$ with the $NO_y$ correction factor from eq. (5) applied in green.

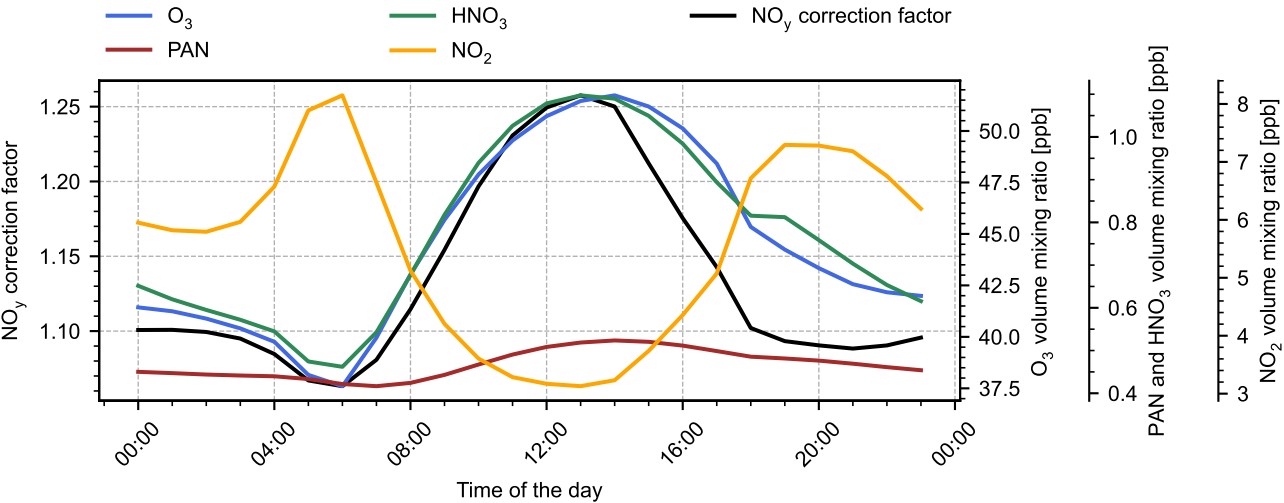

**Figure 4.** Average diurnal cycle of the $NO_y$ correction factor (black), simulated $O_3$ (blue), $HNO_3$ (green), PAN (red), and $NO_2$ (orange). The simulation results are taken from the base run S-YSU and given as volume mixing ratios in order to conform to eq. (5).

as well as the cross sensitive compounds PAN and $HNO_3$ are produced photochemically. Our results are therefore in good agreement with Poraicu et al. (2023), who obtained a correction factor of $\sim 1.2$ at noontime.

We now compare these baseline results to the results obtained from the alternative model runs, as described in sect. 2.2, 2.3, and 2.4. Figure 5 shows the diurnal cycle of simulated and observed surface $NO_2$ for all simulation variants, with and without $NO_y$ correction applied. All model variants, except for S-YSU-TP (the run with tuned temporal emission profiles), show generally better agreement to the noontime observations, when the $NO_y$ correction is applied. The diurnal cycle produced by S-MYJ is similar to that of S-YSU, which has been discussed earlier, with the exception of a slightly stronger noontime low

bias (-23.9 % without $NO_y$ correction, -4.5 % with $NO_y$ correction). S-BouLac, however, produces noontime values similar to S-YSU with highly reduced bias at nighttime (+16.1 % without $NO_y$ correction, +23.8 % with $NO_y$ correction). This motivates to investigate further, how the diurnal cycle of surface $NO_2$ is influenced by the choice of PBL scheme.

Figure 6a shows the diurnal cycle of boundary layer height (BLH) in the YSU, MYJ, and BouLac scheme, averaged over all model cells which contain a UBA station (i.e. the same cells that were used to produce Fig. 5). The highest noontime BLHs

are obtained from the BouLac scheme, while MYJ yields the lowest noontime values. Note that at nighttime, this relationship is inverted. Nonetheless, the simulation run S-BouLac produced much lower nighttime surface $NO_2$ than S-YSU and S-MYJ (see Fig. 5). In a well-mixed boundary layer, one would expect the opposite, seeing that the trace gases inside a more shallow boundary layer are confined into a smaller total volume. An explanation to these results is found by examination of the mixing coefficients $k_h$ in the lowest two model layers, as shown in Fig. 6b: At nighttime, all three schemes produce average mixing

coefficients smaller than $1\,\mathrm{m^2\,s^{-1}}$ in the lowest layer. As described in sect. 2.3, the mixing coefficients are clipped to $1\,\mathrm{m^2\,s^{-1}}$ over rural regions and $2\,\mathrm{m^2\,s^{-1}}$ over urban regions, i.e. the average nighttime mixing strength is then identical for all three

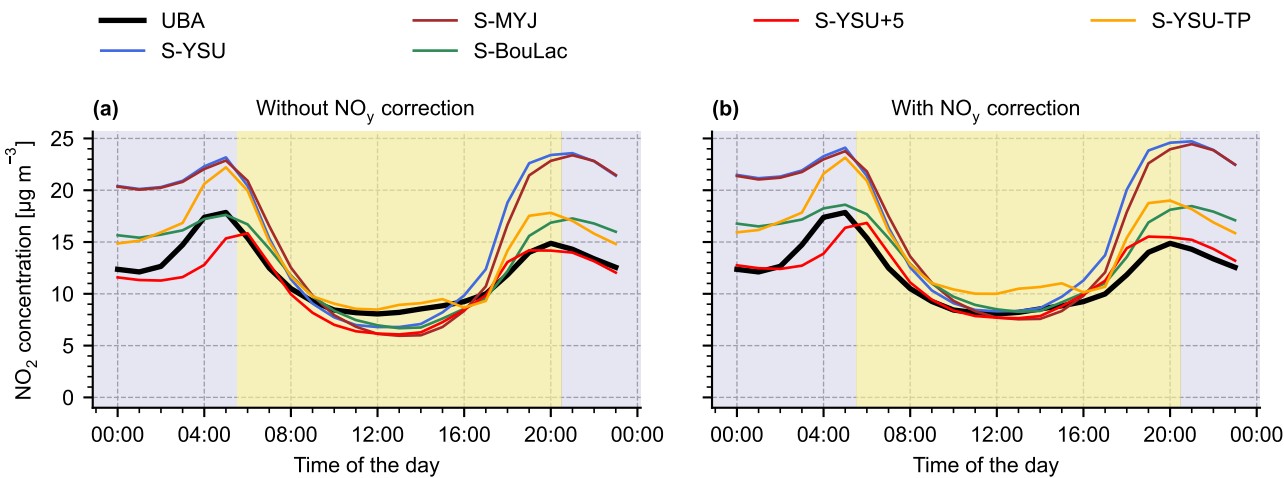

**Figure 5.** Average diurnal cycles of modelled and observed surface $NO_2$ at background stations for the different model variants described in sect. 2.2, 2.3, and 2.4. Day- and nighttime are indicated by yellow and blue background color, respectively. Subplot **(a)** shows the model results without $NO_y$ correction, while subplot **(b)** shows model results with $NO_y$ correction applied.

schemes. One layer above, however, the clipping threshold for $k_h$ is exceeded by BouLac and MYJ. As a consequence, the $NO_x$ emissions at the surface are more efficiently diluted by vertical transport, hence nighttime $NO_2$ concentrations in S-BouLac are significantly lower. The explanation is equally applicable to the daytime observations, where for example, S-MYJ shows the lowest surface $NO_2$ concentrations and the highest mixing coefficients out of all three runs. Figure 7 demonstrates, that the mixing coefficients quickly reach far higher values in higher layers during daytime (shown here for the YSU scheme). In that case, clipping the mixing coefficient to $1\,\mathrm{m^2\,s^{-1}}$, $2\,\mathrm{m^2\,s^{-1}}$, or even $5\,\mathrm{m^2\,s^{-1}}$ (as in S-YSU+5) has no effect, as opposed to nighttime, when the mixing coefficients are mostly smaller than $1\,\mathrm{m^2\,s^{-1}}$. As a result, the diurnal cycle of surface $NO_2$ in the simulation run S-YSU+5 (see Fig. 5) shows significantly reduced nighttime bias (-11.1 % without $NO_y$ correction, as opposed to +53.1 % in S-YSU, and -3.1 % with $NO_y$ correction, as opposed to -60.5 % in S-YSU) and noontime biases comparable to those of S-YSU and S-MYJ (-23.5 % without $NO_y$ correction and -4.4 % with $NO_y$ correction). S-YSU+5 outperforms all other simulation runs at daytime, and nighttime. It should be emphasized that this analysis is based on averaged values of the mixing coefficient. Due to the spatial and temporal variability of $k_h$, individual model cells may be subject to the clipping procedure during daytime as well, explaining the slightly lower noontime $NO_2$ concentrations observed in S-YSU+5. Figure A2 in Appendix A shows a corresponding histogram of noontime mixing coefficients. Similarly, regions at the top of the PBL are affected by the clipping, but play no significant role in our evaluation study.

Next we discuss the results of the simulation run S-YSU-TP with tuned temporal emission profiles. Figure 5a shows, that without $NO_y$ correction, the corresponding model run S-YSU-TP outperforms S-YSU, S-MYJ, and S-BouLac. The model biases are reduced to +5.2 % and -20.4 % at day- and nighttime respectively. As such, it also outperforms S-YSU+5 at noon-

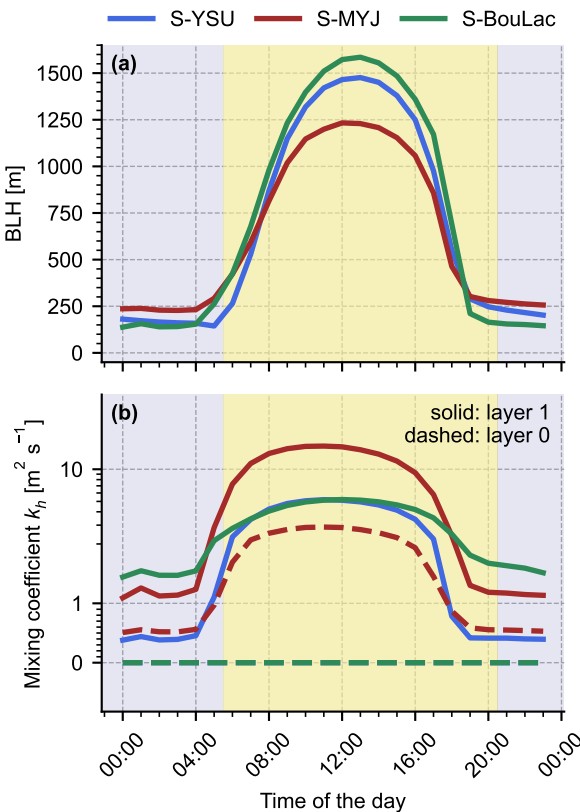

**Figure 6.** Diurnal cycles of boundary layer height **(a)** and mixing coefficient $k_h$ **(b)**, averaged over all model cells which contain a UBA station (i.e. the same cells that were used to produce Fig. 5). Note that the y-axis in **(b)** has a linear scale from $0 \, \text{m}^2 \, \text{s}^{-1}$ to $1 \, \text{m}^2 \, \text{s}^{-1}$, and a logarithmic scale above. Day- and nighttime are indicated by yellow and blue background color, respectively.

time. With $NO_y$ correction applied, S-YSU-TP becomes the worst-performing model run at noontime, with a high model bias of +24.1 %, while still outperforming S-YSU, S-MYJ and S-BouLac at nighttime. Nonetheless, S-YSU-TP is overall outperformed by S-YSU+5. The optimization of temporal profiles alone can therefore not lead to satisfying results. One way to achieve better model performance in S-YSU-TP would be to scale the total emissions (i.e. through the monthly temporal profiles). This, however, causes severe problems in the comparison to the other observational datasets, which indicates that this approach would be inappropriate. Another option would be to run a combined optimization of $k_h$ and the temporal profiles, which could improve the performance of S-YSU+5 further.

So far, the model evaluation was centered around $NO_2$. Figure 8 shows the comparison of diurnal cycles for $NO_x$ instead. Overall, similar qualitative results are obtained, but there are relevant differences to be discussed: The morning NO peak contributes significantly to the morning $NO_x$ concentration and acts in favour of the original model runs S-YSU and S-MYJ, as well as S-YSU-TP, reducing their overall bias. The peak is caused by photolysis of $NO_2$ and the nighttime reservoir species

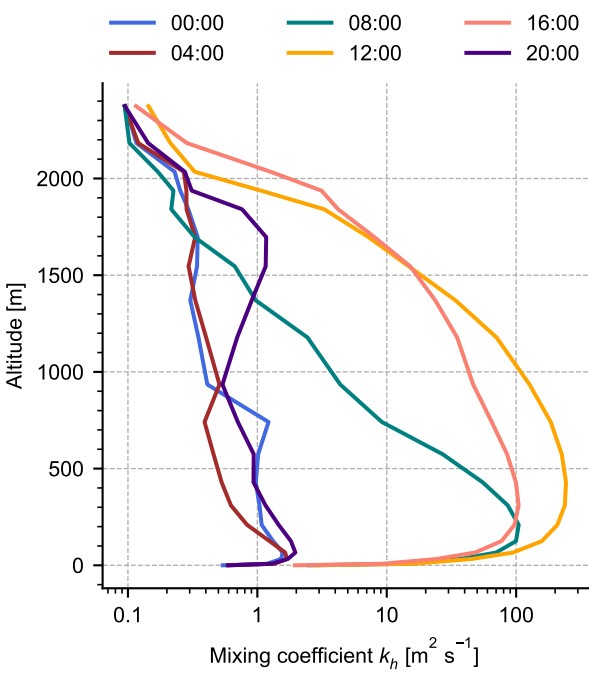

**Figure 7.** Average mixing coefficient profiles from the YSU PBL scheme at different hours of the day.

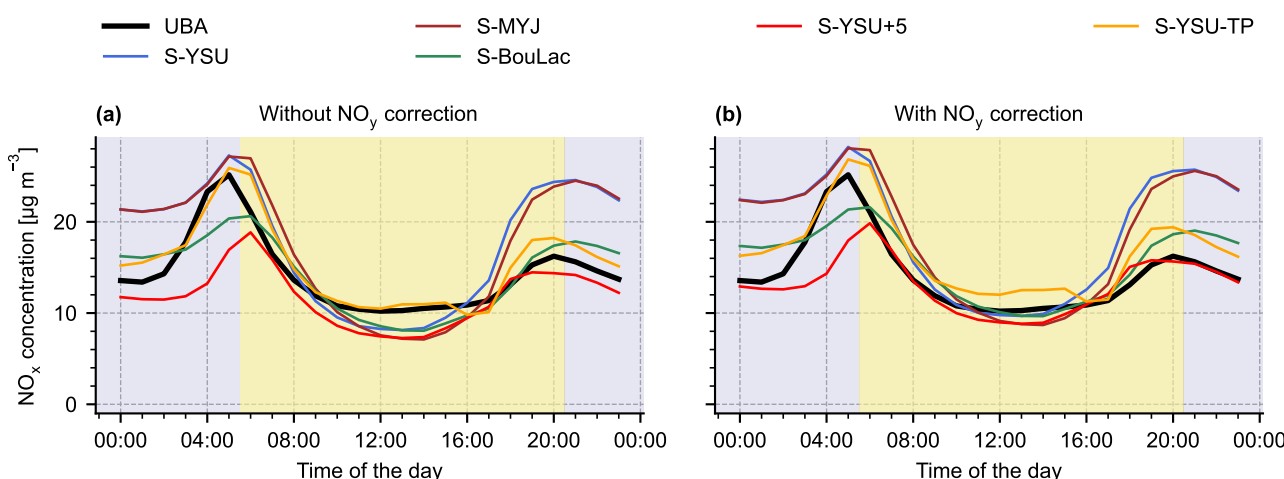

**Figure 8.** Like Fig. 5, but for $NO_x$ instead of $NO_2$.

(such as $NO_3$ and $N_2O_5$). The model, however, obtains the high morning concentrations in addition to the chemical production by underestimating vertical mixing (as shown earlier in the comparison between model runs), i.e. for a wrong reason. Based on the mean daytime bias (-3.0 %), S-YSU+5 still outperforms the other runs. At noontime, however, S-YSU+5 shows a slightly larger low bias (-12.1 %) than the other runs, and it is outperformed by S-BouLac and S-YSU-TP during the nighttime, with a bias of -16.3 %. The results of Fig. 8 suggest that the modelled $NO_x$ concentrations can be further optimized by slightly reducing the lower mixing threshold starting from a value of 5 $m^2\ s^{-1}$. However, this would worsen the agreement with the observed $NO_2$ concentrations from Fig. 5 as well. Another noteworthy observation is the general overestimation of $NO_2/NO$ ratios throughout the day (see Fig. 3), which was observed across all model runs. It was tested in further control runs (not shown here), that scaling of $O_3$ boundary conditions and VOC emissions barely impacts the $NO_2/NO$ ratio of the model. Therefore, it is unlikely that the overestimation of $NO_2/NO$ ratios relates to NO oxygenation alone. Instead, more complex processes in the nighttime $NO_x$ chemistry (see e.g. Knote et al., 2015), or the lack of daytime HONO chemistry could be possible causes. Finally, $NO_2/NO$ ratios could be influenced by the choice of $NO_x$ speciation profiles. In this study, we have used a uniform speciation based on literature values, but it can be assumed that in reality, speciation differs between individual emission sectors (see e.g. Wild et al. (2017); Jimenez et al. (2000); Richmond-Bryant et al. (2017); Costantini et al. (2016)).

The results of this section are summarized in Tables 3 and 4, where only $NO_y$-corrected results are shown.

**Table 3.** Statistical summary of the performance of the five model runs at noontime (12 PM), daytime (6 AM - 8 PM) and nighttime (9 PM - 5 AM). All numbers refer to simulated $NO_2$ at the location of the background UBA stations.

| Simulation | sim. $NO_2$ [$\mu g\ m^{-3}$] | | | Bias | | | RMSE [$\mu g\ m^{-3}$] | | | $R$ | | |
|---|---|---|---|---|---|---|---|---|---|---|---|---|
| | noon | day | night | noon | day | night | noon | day | night | noon | day | night |
| S-YSU | 8.3 | 13.8 | 22.7 | +3.1 % | +30.9 % | +60.5 % | 1.6 | 5.2 | 9.5 | 0.75 | 0.81 | 0.50 |
| S-MYJ | 7.7 | 13.3 | 22.5 | -4.5 % | +26.1 % | +59.4 % | 1.6 | 4.8 | 9.2 | 0.76 | 0.84 | 0.58 |
| S-BouLac | 8.5 | 12.0 | 17.5 | +5.5 % | +13.7 % | +23.8 % | 1.7 | 2.9 | 5.4 | 0.70 | 0.85 | 0.63 |
| S-YSU+5 | 7.7 | 11.1 | 13.7 | -4.4 % | +5.2 % | -3.1 % | 1.6 | 2.3 | 4.1 | 0.74 | 0.83 | 0.43 |
| S-YSU-TP | 10.0 | 13.2 | 18.1 | +24.1 % | +25.0 % | +27.7 % | 2.6 | 3.7 | 5.4 | 0.79 | 0.83 | 0.60 |

Observed mean at noontime: 8.1 $\mu g\ m^{-3}$. Observed mean at daytime: 10.5 $\mu g\ m^{-3}$. Observed mean at nighttime: 14.1 $\mu g\ m^{-3}$.

## 3.2 Comparison of $NO_2$ vertical column densities

As a second diagnostic, we compare simulated $NO_2$ VCDs to observations of the TROPOMI satellite instrument (see Veefkind et al., 2012). The satellite overpass occurs daily, typically at 13:30 local time in central Europe, with a pixel size of 3.5 km × 5.5 km in NADIR viewing geometry. TROPOMI measures spectra of backscattered sunlight from earth's surface, from which tropospheric slant column densities (SCDs) are computed using Differential Optical Absorption Spectroscopy (DOAS, see Platt and Stutz, 2008). Tropospheric VCDs are obtained from tropospheric SCDs via

$$VCD_{trop} = \frac{SCD_{trop}}{AMF_{trop}} \tag{6}$$

**Table 4.** Statistical summary of the performance of the five model runs at noontime (12 PM), daytime (6 AM - 8 PM) and nighttime (9 PM - 5 AM). All numbers refer to simulated $NO_x$ at the location of the background UBA stations.

| Simulation | sim. $NO_x$ [$\mu g\ m^{-3}$] | | | Bias | | | RMSE [$\mu g\ m^{-3}$] | | | $R$ | | |
|---|---|---|---|---|---|---|---|---|---|---|---|---|
| | noon | day | night | noon | day | night | noon | day | night | noon | day | night |
| S-YSU | 9.8 | 15.7 | 24.2 | -4.2 % | +22.4 % | +43.7 % | 2.0 | 5.4 | 9.4 | 0.75 | 0.77 | 0.54 |
| S-MYJ | 9.1 | 15.4 | 24.1 | -10.6 % | +20.1 % | +43.4 % | 2.3 | 5.3 | 9.2 | 0.74 | 0.81 | 0.60 |
| S-BouLac | 10.1 | 13.8 | 18.5 | -1.1 % | +7.2 % | +9.7 % | 2.1 | 3.3 | 6.1 | 0.67 | 0.82 | 0.59 |
| S-YSU+5 | 9.0 | 12.5 | 14.1 | -12.1 % | -3.0 % | -16.3 % | 2.2 | 2.8 | 6.8 | 0.73 | 0.81 | 0.48 |
| S-YSU-TP | 12.0 | 15.2 | 18.9 | +17.6 % | +18.2 % | +12.5 % | 2.8 | 3.9 | 5.5 | 0.78 | 0.83 | 0.68 |

Observed mean at noontime: 10.2 $\mu g\ m^{-3}$. Observed mean at daytime: 12.9 $\mu g\ m^{-3}$. Observed mean at nighttime: 16.8 $\mu g\ m^{-3}$ .

where $AMF_{trop}$ stands for the tropospheric air mass factor. The tropospheric AMF depends on a number of atmospheric and surface conditions and assumptions, e.g. cloud properties, the stratospheric column, surface albedo, and the a priori $NO_2$ profile. In the TROPOMI retrieval, the $NO_2$ a priori profiles are taken from the TM5 global CT model (see Krol et al. (2005); Williams et al. (2017)), with a spatial resolution of $\sim 1° \times 1° \approx 110\ km \times 70\ km$ where our simulation domain D2 is located. A detailed description of the retrieval algorithm can be found in van Geffen et al. (2022).

Simulated VCDs are obtained by vertical integration of box VCDs (= partial columns) from WRF-Chem. The WRF-Chem output is interpolated to the vertical grid of the TM5 model. The box VCDs are computed by multiplying the $NO_2$ concentration within each model grid cell with its vertical extent. The partial columns are then summed up:

$$VCD_{sim} = \sum_{l<l_{tp}} c_l \cdot \Delta h_l \tag{7}$$

where $VCD_{sim}$ denotes the simulated VCD, $l$ the TM5 model layer index, $l_{tp}$ the tropopause layer index, $c_l$ the $NO_2$ concentration in layer $l$, and $\Delta h_l$ the vertical extent of layer $l$. Then, pairs of simulated and observed VCDs are obtained by interpolating the WRF-Chem data to the horizontal TROPOMI grid in space and time.

In order to make a representative comparison between simulated and observed VCDs, the following further aspects are taken into consideration:

1. TROPOMI is not equally sensitive to all layers of the troposphere. This circumstance is described by the averaging kernels (AKs, see van Geffen et al., 2022). Furthermore, the AMF depends on the $NO_2$ a priori profiles provided by the TM5 model. It has been demonstrated that replacing TM5 a priori profiles with profiles from high-resolution RCT models improves the VCD retrieval significantly (see e.g. Ialongo et al. (2020); Tack et al. (2021); Liu et al. (2021)).

We incorporate the AKs and our simulated high-resolution $NO_2$ a priori profiles by computing corrected observed VCDs following Eskes et al. (2019):

$$VCD_{obs,\ corr} = VCD_{obs} \cdot \frac{AMF_{trop}}{AMF} \cdot \frac{\sum_{l<l_{tp}} c_l \cdot \Delta h_l}{\sum_{l<l_{tp}} c_l \cdot \Delta h_l \cdot A_l} \tag{8}$$

where $VCD_{obs}$ denotes the observed $NO_2$ VCD, $AMF_{trop}$ the tropospheric AMF, and $A$ the averaging kernel vector.

2. Each observed VCD has an associated $qa$-value that describes the retrieval quality on a range from 0 (bad) to 1 (good) (see van Geffen et al., 2022). $qa$-values of $\geq 0.75$ also require the cloud radiance fraction (crf) to be smaller than 0.5, which effectively acts as a cloud filter. The effect of clouds on the measurement process depends on the height of the cloud layer: High clouds can shield off lower layers of the atmosphere, decreasing the $NO_2$ sensitivity to almost zero below the cloud layer. Vice versa, cloud layers directly below $NO_2$ layers can enhance the observed $NO_2$ absorption due to the increased cloud albedo. If cloud layers and $NO_2$ layers are at the same altitude, $NO_2$ absorption can be increased by an enhanced light path due to multiple scattering. The influence of clouds on satellite measurements of $NO_2$ has been discussed e.g. in Martin et al. (2002), Kokhanovsky and Rozanov (2008), and Liu et al. (2021). We apply a $qa$-filter that removes all observations with $qa < 0.75$, as recommended by Eskes et al. (2019).

Figure 9 shows the result of our comparison in the form of monthly mean values (i.e. the total average over May 2019) for the base simulation run S-YSU. Subfigures a-e show the comparison between simulated $NO_2$ VCDs and TROPOMI observations using the TM5 $NO_2$ a priori profiles. Strong enhancements occur in west Germany (Ruhr region) and along the Rhine river. Moderate enhancements occur over logistical hotpots (e.g. the port of Hamburg with a nearby coal power plant), and larger cities (e.g. Berlin and Munich; see Fig. 11 for an overview of the geographical regions mentioned here). Subfigure c shows the difference between simulated and observed VCDs and reveals the model's tendency to overestimate the observed VCDs. The overestimation is close to zero in rural regions but reaches up to $10^{16}$ molec. cm$^{-2}$ over hotspot regions. The simulated and observed VCDs correlate with an $R$-value of 0.85, an RMSE of $1.02 \cdot 10^{15}$ molec. cm$^{-2}$, and a bias of $+21.1$ %. A linear fit through the point cloud (subfigure e) yields a slope of 1.28 and an intercept of $-0.01 \cdot 10^{15}$ molec. cm$^{-2}$. Subfigures f-j show the same comparison, but with recomputed airmass factors according to eq. (8). The comparison improves, with an $R$-value of 0.87, an RMSE of $0.85 \cdot 10^{15}$ molec. cm$^{-2}$, a bias of $+6.7$ %, a slope of 1.16 and an intercept of $-0.02 \cdot 10^{15}$ molec. cm$^{-2}$.

Subfigures k-o show the spatial distribution of simulated and observed surface $NO_2$ concentrations from the UBA background stations. Model overestimations of $NO_2$ VCD and surface concentration occur in similar geographic regions, particularly in the Ruhr region, along the Rhine river, and the port of Hamburg. This hints towards a possible overestimation of $NO_x$ emissions in these regions. Figure 10 presents further comparisons of simulated and observed $NO_2$ VCDs for different simulation runs. Here, we only show comparisons using reprocessed observations according to eq. (8). Table 5 summarizes the results of this section. The main observation to be made here is that the high sensitivity of the simulated surface concentrations (see sect. 3.1) to the PBL scheme and vertical mixing strength is not found in the comparison for column densities, as expected for a trace gas column. Even the base run S-YSU shows good results, with a slope of 1.16 and an RMSE of $0.85 \cdot 10^{15}$ molec. cm$^{-2}$. S-YSU+5 achieves the lowest RMSE ($0.80 \cdot 10^{15}$ molec. cm$^{-2}$), and slightly better slope of 1.12. In addition, the overestimation over West Germany appears to be reduced compared to all other runs. Overall, S-YSU, S-MYJ, S-BouLac and S-YSU+5 perform quite similarly with respect to the statistical diagnostics shown here (with the exception of S-BouLac, showing a significantly higher RMSE of $0.93 \cdot 10^{15}$ molec. cm$^{-2}$). This is expected, seeing that in first order, the tropospheric column is a measure of total $NO_2$, which may vary only slightly across these simulation runs. S-YSU-TP shows

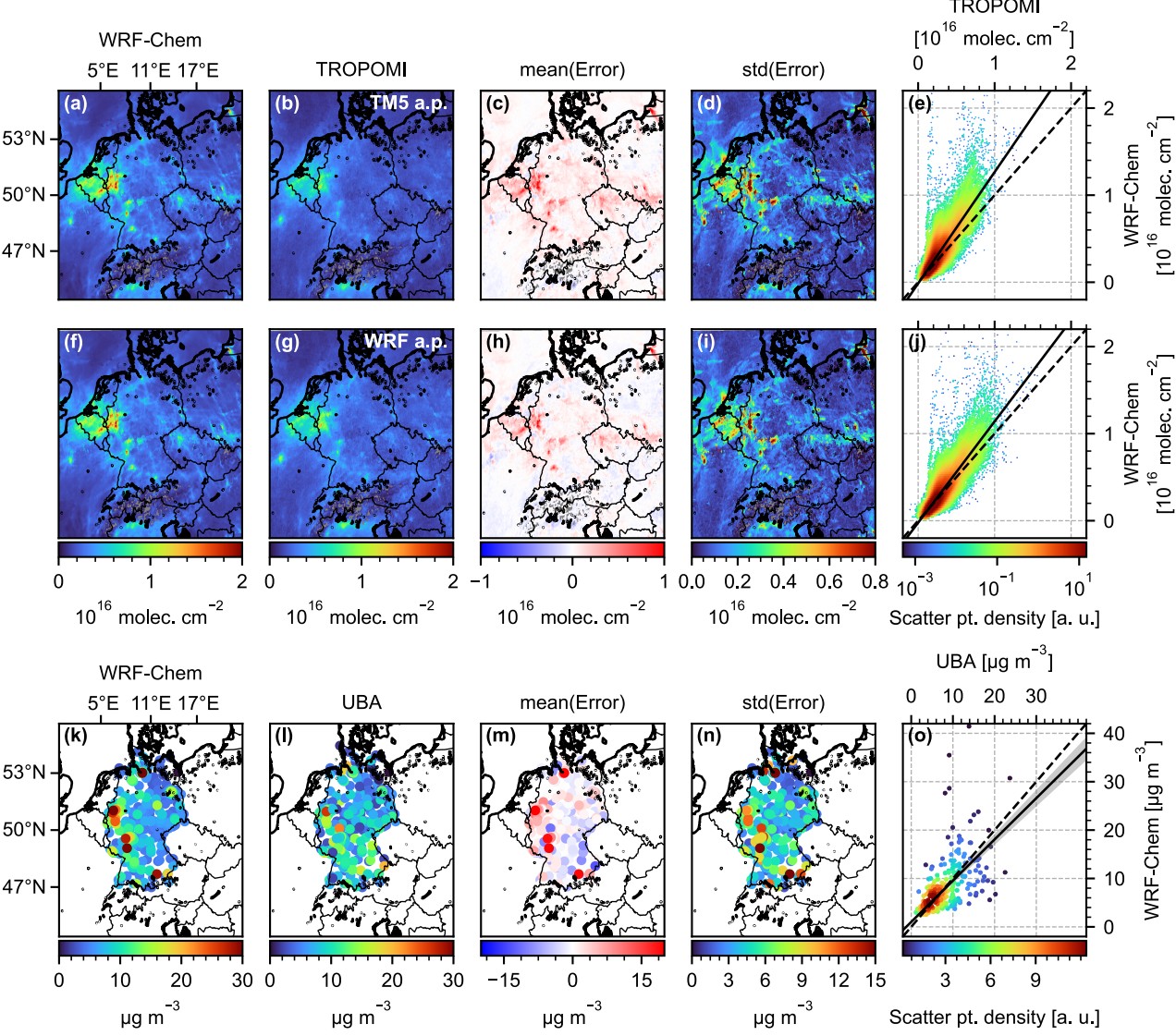

**Figure 9.** Comparison of simulated and observed $NO_2$ VCDs **(a)**-**(j)** and surface $NO_2$ concentrations **(k)**-**(o)** for the base model run S-YSU. The first row **(a)**-**(e)** shows the comparison to the original observations using low-resolution TM5 a priori $NO_2$ profiles. The second row **(f)**-**(j)** shows the comparison with reprocessed observations using high-resolution WRF-Chem a priori profiles and averaging kernels. All satellite observations were restricted to cases with $qa \geq 0.75$. The third row **(k)**-**(o)** shows a comparison between simulated and observed surface $NO_2$ concentrations from "background" stations.

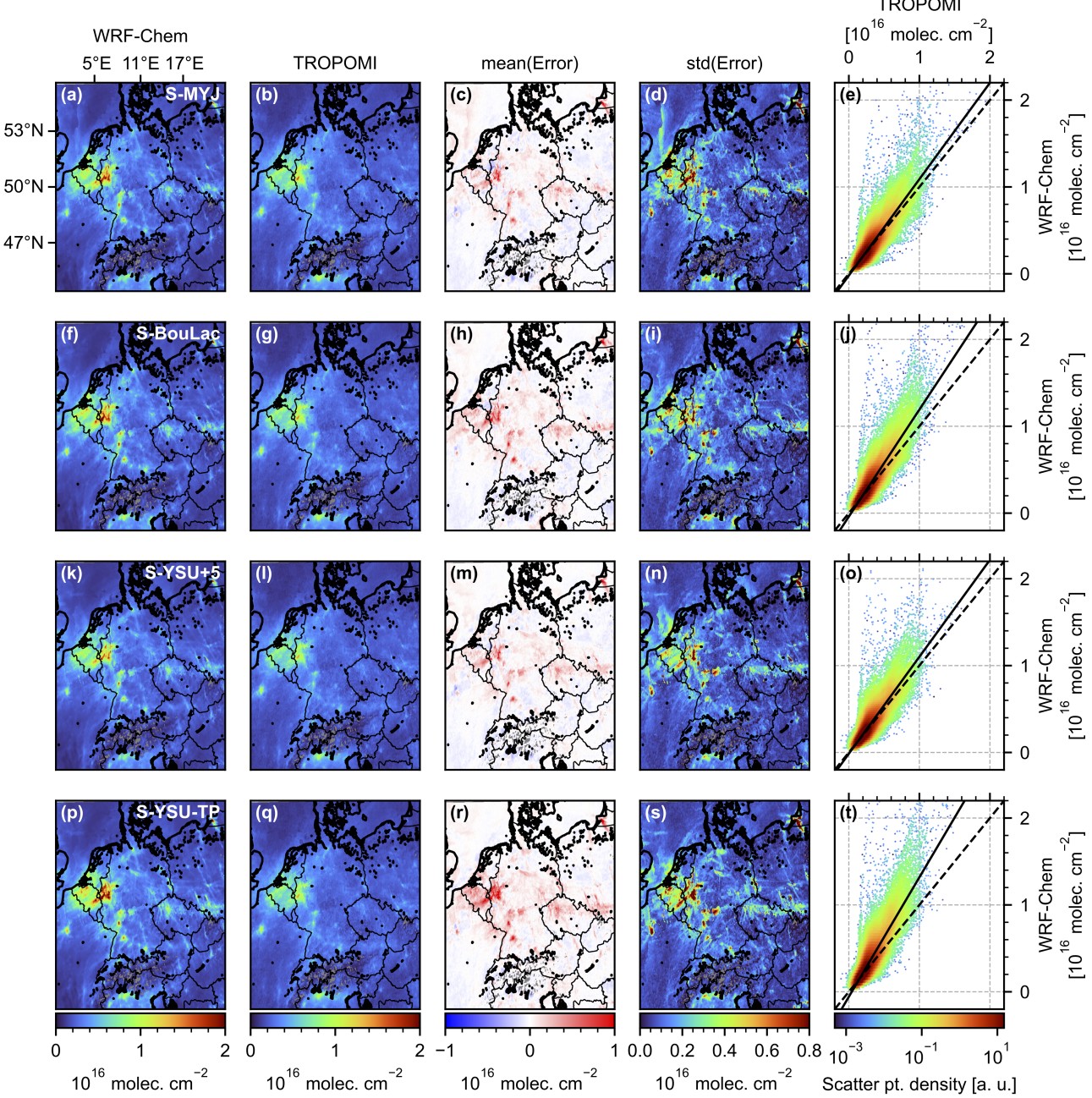

**Figure 10.** Further comparisons of simulated and observed NO$_2$ VCDs for the remaining simulation runs S-MYJ, S-BouLac, S-YSU+5, and S-YSU-TP. The comparison for each model run is in direct analogy to Fig. 9f-j.

**Table 5.** Statistical summary of the results shown in Fig. 9 and 10

| Simulation run | Mean (sim.) | Mean (obs.) | Slope | Intercept | RMSE | Bias | $R$ |
|---|---|---|---|---|---|---|---|
| S-YSU (TM5 a priori profiles) | 2.56 | 2.11 | 1.28 | -0.01 | 1.02 | +21.1 % | 0.85 |
| S-YSU | 2.56 | 2.40 | 1.16 | -0.02 | 0.85 | +6.7 % | 0.87 |
| S-MYJ | 2.60 | 2.53 | 1.11 | -0.02 | 0.83 | +3.0 % | 0.88 |
| S-BouLac | 2.64 | 2.47 | 1.23 | -0.04 | 0.93 | +6.9 % | 0.89 |
| S-YSU+5 | 2.51 | 2.38 | 1.12 | -0.01 | 0.80 | +5.5 % | 0.87 |
| S-YSU-TP | 2.76 | 2.42 | 1.38 | -0.06 | 1.14 | +13.8 % | 0.87 |

Means and RMSE are given in units of $[10^{15} \ \mathrm{molec. \ cm}^{-2}]$

significantly larger simulated VCDs, resulting in a bias of +13.8 % and a RMSE of $1.14 \cdot 10^{15}$ molec. cm$^{-2}$ (the worst out of all runs). Figure 10r shows, that S-YSU-TP overestimates the $NO_2$ VCDs over West Germany, the Netherlands, and Belgium
more strongly than the other runs, while not performing noticeably better in any other part of the domain. The combined results of Tables 3, 4, and 5 favor S-YSU+5, the run with enhanced vertical mixing, as the best setup.

### 3.3  Comparison of $NO_2$ concentration profiles

As a third diagnostic we compare modelled $NO_2$ concentration profiles to profiles obtained from MAX-DOAS (Multi-Axis Differential Optical Absorption Spectroscopy) measurements. In contrast to the data shown in sect. 3.1 and 3.2, profile com-
parison allows for assessment of the model's capability to capture vertical distributions of $NO_2$. MAX-DOAS measurements use the DOAS principle (see Platt and Stutz, 2008) to obtain trace gas differential slant column densities (dSCDs) at different elevation angles. By application of an inversion algorithm a discretized concentration vector $c$ is obtained, whose entries denote the target gas concentration in different atmospheric layers. An overview of different inversion algorithms can be found in Frieß et al. (2019). We use data from the FRM4DOAS network (see Fayt et al., 2021) which applies the Mexican MAX-DOAS
fit (MMF, see Friedrich et al., 2019) and the Mainz Profile Algorithm (MAPA, see Beirle et al., 2019). While MMF is based on optimal estimation (see Rodgers, 2000), MAPA uses Monte Carlo simulation in order to determine profile shape parameters which combine (possibly lifted) box profiles and exponential profiles. Purely exponential concentration profiles, however, can not be obtained from the current version of MAPA.

Five MAX-DOAS instruments are operated within our simulation domain D2: Mainz (Germany), Bremen (Germany), Hei-
delberg (Germany), De Bilt (Netherlands), and Uccle (Belgium). Figure 11 shows the locations of these stations. A single station typically yields 2-4 $NO_2$ profile measurements per hour. In order to compare simulated and observed profiles, the WRF-Chem dataset is interpolated to the geolocations and measurement times of the MAX-DOAS instruments. The uncertainty of the simulated $NO_2$ profiles is obtained as the standard deviation of the surrounding eight WRF-Chem grid cells.

The MMF inversion algorithm provides averaging kernels (AKs), represented by a $(h \times h)$ matrix, where $h$ is the number
of atmospheric layers considered. Here, $h = 20$, comprising retrieval altitudes up to 4 km. Using the AKs, the MAX-DOAS

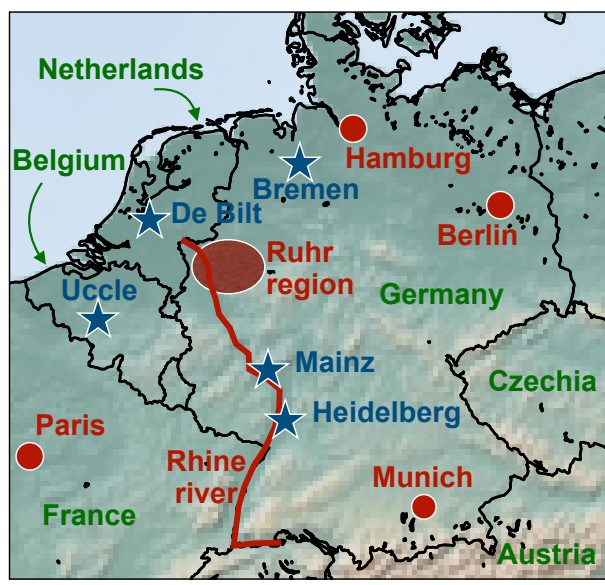

**Figure 11.** Part of central Europe covering Germany, Belgium and Netherlands with important geographical regions marked in red. The blue stars show the locations of the MAX-DOAS measurements shown in sect. 3.3.

measuring sensitivity to different altitudes can be applied to the simulated profiles. For this purpose the simulated $NO_2$ profiles are vertically interpolated to the layers of the MAX-DOAS retrieval grid. Then, the AKs are applied via

$$c_{\text{sim, corr}} = \mathbf{A}c_{\text{sim}} + (\mathbb{1} - \mathbf{A})c_{\text{ap}} \tag{9}$$

where $c_{\text{sim, corr}}$ is the corrected simulated profile, $\mathbf{A}$ the averaging kernel matrix, $c_{\text{sim}}$ the original simulated profile, $\mathbb{1}$ the unity matrix, and $c_{\text{ap}}$ the a priori profile see (see Rodgers and Connor, 2003). MAPA neither provides AKs, nor depends on an a priori profile. All profiles flagged as "erroneous" by MAPA were dismissed from the evaluation.

Figure 12 shows averaged $NO_2$ profiles from the MAX-DOAS station Mainz, Germany, in the time window from 11 AM to 02 PM for a selection of individual days, as well as the corresponding modelled profiles from the base run S-YSU. The aim is to give an overview of the variety of observed and modelled profile shapes. Additionally, the average $NO_2$ surface concentrations measured by UBA stations within a radius of 5 km of the MAX-DOAS instrument are drawn as colored scatter markers at 0 m altitude.

Figure 12a shows a typical scenario: On this day (03 May 2019) observation and simulation show comparably good agreement. All three datasets (MMF, MAPA, and WRF-Chem) yield profiles of similar shape. Above an altitude of $\sim 2$ km the profiles quickly approach zero, which is characteristic for the transitioning regime between the PBL and the free troposphere. At low altitudes of 0-150 m the simulated profile has a strong exponential tail, typical for the surface layer into which most $NO_x$ emissions are injected (the lowest $\sim 30$ m). Furthermore, the simulated profiles show good agreement with the measurements

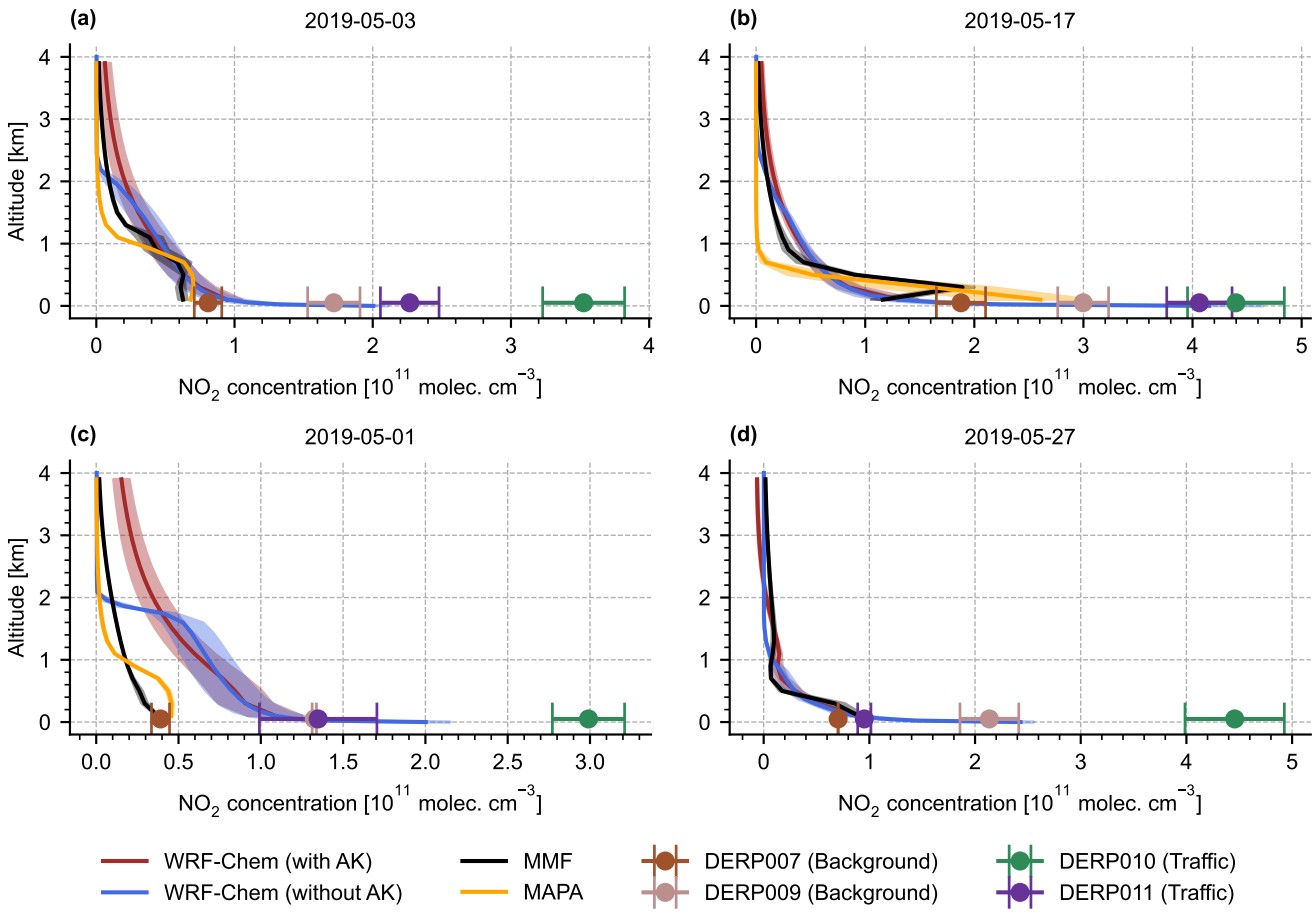

**Figure 12.** Comparison of average $NO_2$ profiles for Mainz around noontime (11 AM - 02 PM) for four exemplary days: **(a)** 03 May 2019, **(b)** 17 May 2019, **(c)** 01 May 2019, **(d)** 27 May 2019. Additionally, average $NO_2$ surface concentrations measured by UBA stations within a radius of 5 km of the MAX-DOAS instrument are drawn as colored scatter markers at 0 m altitude. The WRF-Chem results are taken from the base run S-YSU.

of the nearby UBA in-situ stations. High vertical resolution turns out to be crucial for the comparison to surface observations: In the FRM4DOAS dataset, MAPA and MMF were operated with a vertical resolution of 200 m. In a direct comparison, the concentrations obtained from MAPA or MMF in the lowest layer (0-200 m) underestimate the in-situ observations at the surface because of the limited vertical resolution of the MAX-DOAS measurements and the rather coarse retrieval grid used by FRM4DOAS. Meanwhile, the lowest layer of our WRF-Chem simulation only spans 0 - 8 m, which allows for a much more representative comparison.

Subfigure 12b shows a day on which an elevated $NO_2$ layer was detected by MMF. This is characterized by strongly enhanced $NO_2$ concentrations at higher altitudes (here at $\sim 500$ m). Elevated layers are typically caused by elevated emissions, e.g. from a power plant stack at a few hundred meters height. Additionally, transport events which advect $NO_2$ from the surface layer could be the cause. However, no corresponding enhancements can be seen in the simulated $NO_2$ profile. As described in sect. 2.1, we do not use vertical emission profiles in the WRF-Chem simulation. Furthermore, it is possible that the overall spatial resolution of our simulation limits its ability to model comparable elevated trace gas abundances. With a resolution of 3 km × 3 km × 100 m at $\sim 500$ m altitude, trace gas concentrations are diluted into comparably large grid cell volumes. Lastly, horizontal concentration gradients can be expected near strong emission sources. These can lead to apparently elevated profiles, because the MAX-DOAS profile inversion makes the simplifying assumption of horizontally homogeneous distributions. Comparison of simulated and observed elevated layers is therefore an advanced problem and not further addressed here.

Subfigure 12c shows a day on which the agreement between model and observations was poor, with deviations of > 100 %. Such days are outliers and do not represent the overall quality of the model or the measurements, however, they should not remain unmentioned. Possible reasons for such deviations include e.g. falsely modelled wind directions (affecting trace gas transport) or cloud cover (affecting photolysis).

Subfigure 12d shows a day on which the modelled profile has an extremely steep exponential gradient in combination with a relatively thin boundary layer. At the transition from the boundary layer to the surface layer ($\sim 30$ - 50 m above ground), the simulated profiles show an increase of almost 400 %. Interestingly, on this day (27 May 2019), one of the background stations (DERP009) measured more than twice the surface concentration compared to a nearby traffic station (DERP011). This phenomenon was observed on only one other day (10 May 2019) of the simulated month, on which very similar profile shapes were observed. Furthermore, no MAPA profile is available on this day due to the filtering by error flags. Also, there is good agreement between modelled and observed profiles above the surface layer. It is therefore plausible to assume that the steep $NO_2$ gradient towards the surface is not due to a faulty model, but, on the contrary, indicates the model's ability to capture scenarios in which e.g. vertical mixing is mostly suppressed and large portions of the emitted $NO_2$ remain in the surface layer.

In order to condense the remaining evaluation, we will focus on monthly noontime averages from hereon, i.e. plots of the same structure as shown in Fig. 12, but averaged over the entire simulation period of May 2019 (Fig. 13 and 14). In both figures the right-side panel shows scatter plots of averaged $NO_2$ concentrations (WRF-Chem vs. MMF, due to availability of AKs) at different altitudes.

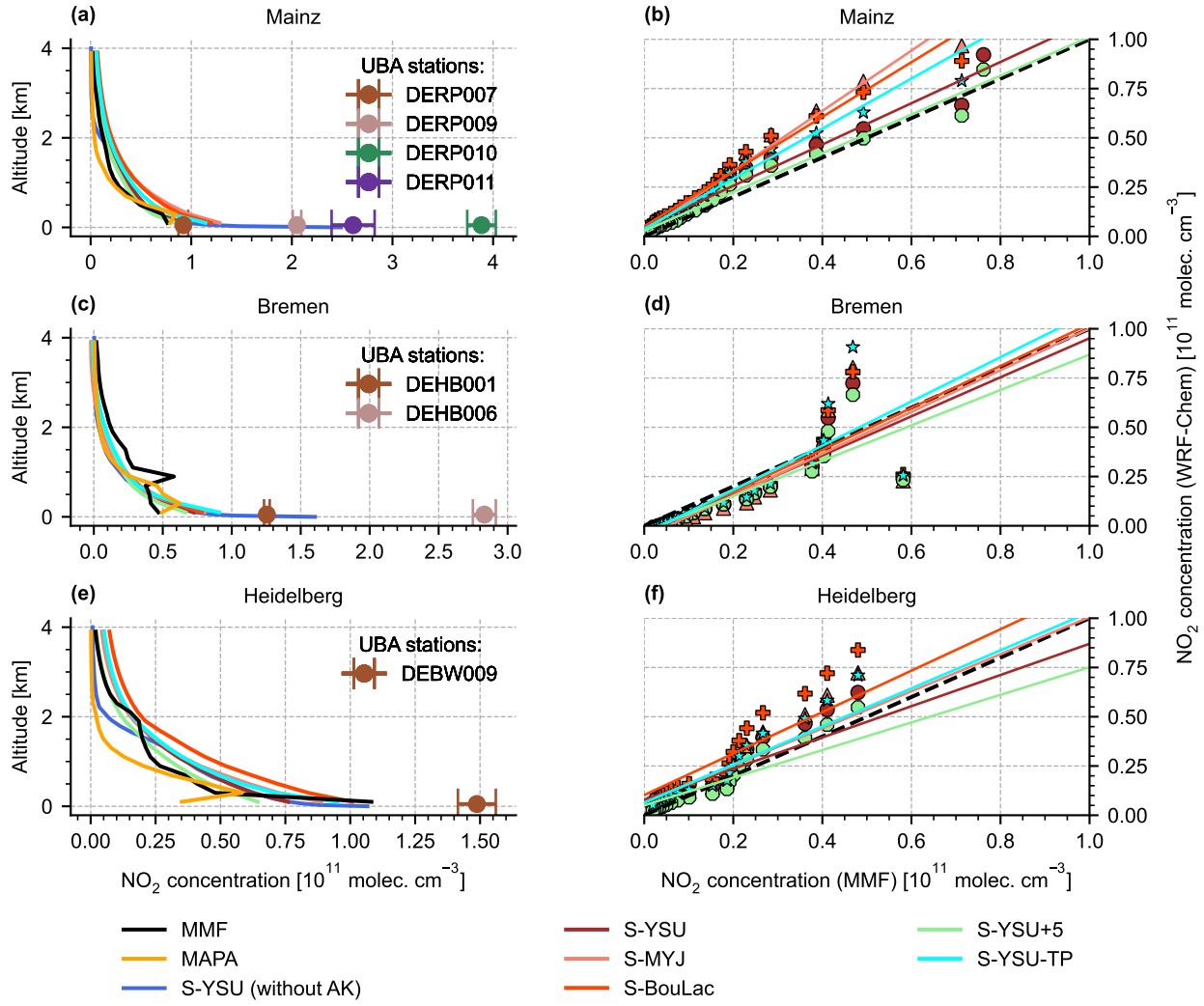

**Figure 13.** Comparison of averaged $NO_2$ profiles for Mainz, Bremen, and Heidelberg around noontime (11 AM - 02 PM). The left column **(a)**, **(c)**, **(e)** shows the $NO_2$ profiles as obtained by MMF, MAPA, and WRF-Chem. The right column **(b)**, **(d)**, **(f)** shows the corresponding scatter plots of averaged $NO_2$ concentrations at different altitudes (here: WRF-Chem vs. MMF).

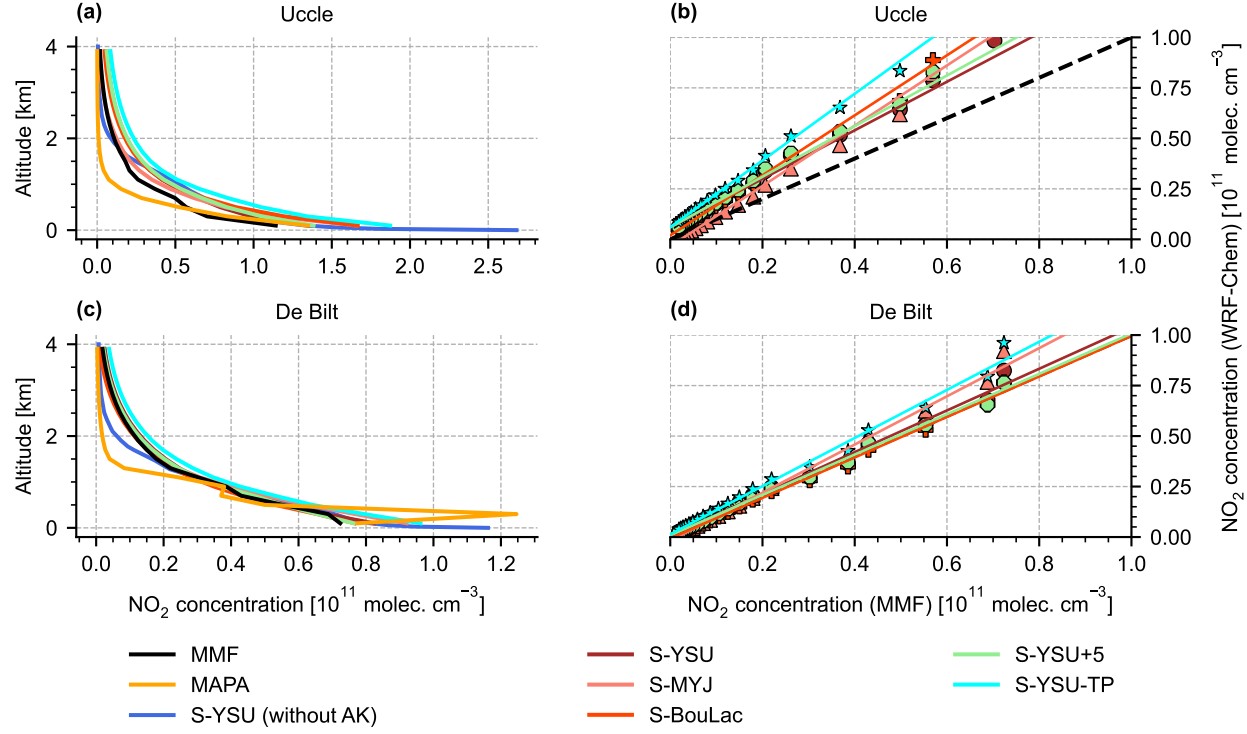

**Figure 14.** Like Fig. 13, but for the stations De Bilt and Uccle.

Figure 13 shows the results of the three German MAX-DOAS locations Mainz, Bremen, and Heidelberg. Overall, the three stations show qualitatively similar profiles. At Bremen (see subfigures c-d), a consistent elevated layer at $\sim 1000$ m altitude was detected, which could be explained by multiple power plants in the near vicinity of the instrument. This has been discussed in detail by Bösch (2018). The scatter plots (subfigures b, d, f) show a strong correlation between simulation and measurements. A quantitative summary of the comparison is found in Table 6. Although the performance of WRF-Chem varies between the five simulation runs, none of them clearly outperforms the others: The simulation run S-YSU+5, which performed best in sect. 3.1 and 3.2 outperforms the other runs in four out of five station with respect to RMSE (exception: Uccle, where S-BouLac has a lower RMSE). However, when considering the bias, S-YSU+5 is outperformed in multiple cases. The simulation run with tuned temporal emission profiles S-YSU-TP produces larger average $NO_2$ profiles than the base run S-YSU. This effect is especially strong in Uccle and De Bilt, and expected due to the strong shift of emissions towards noontime. However, in Mainz and Heidelberg, S-BouLac (and to some extent, S-MYJ) produce even larger $NO_2$ concentrations near ground, which contradicts the general tendencies observed in sect. 3.1 (see Fig. 5). It should be considered, that two exemplary observations of this phenomenon are not significant compared to the hundreds of in-situ measurement stations shown in sect. 3.1. Also, the MAX-DOAS stations are closer to strong emission sources in generally more polluted locations, whereas only background

stations were considered in the comparison to in-situ observations. A temporally resolved version of the comparison is found

**Table 6.** Statistical summary of the results shown in Fig. 13 and 14

| Run and location | Mean (WRF-Chem) | Mean (MMF) | Slope | Intercept | RMSE | Bias | $R$ |
|---|---|---|---|---|---|---|---|
| S-YSU (Mainz) | 0.26 | 0.20 | 1.05 | 0.05 | 0.07 | +27.7 % | 0.98 |
| S-MYJ (Mainz) | 0.34 | 0.20 | 1.53 | 0.02 | 0.18 | +65.0 % | 0.99 |
| S-BouLac (Mainz) | 0.34 | 0.20 | 1.39 | 0.05 | 0.16 | +63.9 % | 0.99 |
| S-YSU+5 (Mainz) | 0.23 | 0.20 | 0.98 | 0.03 | 0.05 | +12.0 % | 0.98 |
| S-YSU-TP (Mainz) | 0.30 | 0.20 | 1.27 | 0.04 | 0.12 | +45.7 % | 0.98 |
| S-YSU (Bremen) | 0.15 | 0.19 | 0.99 | -0.04 | 0.11 | -21.2 % | 0.85 |
| S-MYJ (Bremen) | 0.15 | 0.19 | 1.06 | -0.06 | 0.13 | -24.0 % | 0.82 |
| S-BouLac (Bremen) | 0.16 | 0.19 | 1.06 | -0.04 | 0.12 | -14.4 % | 0.85 |
| S-YSU+5 (Bremen) | 0.14 | 0.19 | 0.90 | -0.03 | 0.11 | -25.3 % | 0.86 |
| S-YSU-TP (Bremen) | 0.17 | 0.19 | 1.12 | -0.04 | 0.14 | -9.7 % | 0.82 |
| S-YSU (Heidelberg) | 0.24 | 0.21 | 0.79 | 0.08 | 0.10 | +16.3 % | 0.92 |
| S-MYJ (Heidelberg) | 0.26 | 0.21 | 0.94 | 0.06 | 0.10 | +24.7 % | 0.93 |
| S-BouLac (Heidelberg) | 0.32 | 0.21 | 1.05 | 0.10 | 0.16 | +54.0 % | 0.92 |
| S-YSU+5 (Heidelberg) | 0.20 | 0.21 | 0.70 | 0.05 | 0.10 | -5.9 % | 0.92 |
| S-YSU-TP (Heidelberg) | 0.27 | 0.21 | 0.97 | 0.06 | 0.09 | +27.2 % | 0.95 |
| S-YSU (Uccle) | 0.34 | 0.23 | 1.20 | 0.06 | 0.12 | +45.4 % | 1.00 |
| S-MYJ (Uccle) | 0.31 | 0.23 | 1.49 | -0.03 | 0.17 | +34.6 % | 1.00 |
| S-BouLac (Uccle) | 0.37 | 0.23 | 1.49 | 0.02 | 0.20 | +57.0 % | 1.00 |
| S-YSU+5 (Uccle) | 0.35 | 0.23 | 1.25 | 0.06 | 0.14 | +50.7 % | 0.99 |
| S-YSU-TP (Uccle) | 0.45 | 0.23 | 1.65 | 0.06 | 0.28 | +90.9 % | 1.00 |
| S-YSU (Debilt) | 0.23 | 0.21 | 1.03 | 0.01 | 0.03 | +6.8 % | 1.00 |
| S-MYJ (Debilt) | 0.24 | 0.21 | 1.19 | -0.02 | 0.05 | +10.3 % | 1.00 |
| S-BouLac (Debilt) | 0.21 | 0.21 | 1.00 | -0.01 | 0.02 | -2.9 % | 1.00 |
| S-YSU+5 (Debilt) | 0.22 | 0.21 | 1.00 | 0.01 | 0.02 | +3.9 % | 1.00 |
| S-YSU-TP (Debilt) | 0.27 | 0.21 | 1.19 | 0.01 | 0.07 | +25.6 % | 1.00 |

Means, intercept, and RMSE are given in units of $[10^{10}\ \mathrm{molec.\ cm^{-3}}]$.

in Fig. A3 - A7. Here, only the lowest 4 layers are shown for each station. The quality of the results varies. In the lowest 0 - 200 m, the shape of the diurnal concentration cycle is reproduced, but offsets do occur (particularly in Bremen). The higher layers show overall much weaker temporal variations, making it increasingly hard to distinguish between true fluctuations of $NO_2$ concentrations and retrieval noise (compare the results obtained from MAPA and MMF for Bremen, Fig. A4). Similarly

to the noon-time evaluation, none of the simulation runs is obviously superior over the others.

## 4 Discussion and conclusion

We have presented new WRF-Chem simulation results with specific focus on $NO_2$. The simulations were run on a domain over central Europe with a spatial resolution of $3\,\mathrm{km} \times 3\,\mathrm{km}$ for the month of May 2019. Over Germany, a new emission inventory from the German Environmental Agency with a resolution of $1\,\mathrm{km} \times 1\,\mathrm{km}$ was used. Outside of Germany, the EDGARv5 emission inventory with a resolution of $11\,\mathrm{km} \times 7\,\mathrm{km}$ was used. The obtained simulation results (surface concentrations, tropospheric columns, and concentration profiles) were compared to observational datasets. On the basis of this comparison, five different model setups were evaluated against each other. The five model runs differed by choice of the PBL scheme, an enhanced mixing approach according to Du et al. (2020), and tuning of the diurnal temporal emission profiles.

In sect. 3.1 modelled $NO_2$ surface concentrations were compared to in-situ measurements. The obtained results were qualitatively very similar to previously published literature (see Kuik et al., 2016; Poraicu et al., 2023), with a model bias of $-15.7\,\%$ at noontime and $+53.1\,\%$ at night. The noontime bias was minimized by correcting for the presumed $NO_y$ cross sensitivities of the reference measurement using the correction term suggested by Lamsal et al. (2008) based on modelled volume mixing ratios of PAN and $HNO_3$, resulting in a noon-time bias of $+3.1\,\%$. The nighttime bias was found to be sensitive to the PBL scheme, and was reduced to $+23.8\,\%$ in a model run with the Bougeault-Lacarrere scheme. The best results were obtained by increasing the lower mixing threshold of the model, as described by Du et al. (2020). As expected from a vertically resolved examination of the model's mixing coefficients, the enhanced mixing approach mainly affected the surface layer of the model at nighttime, where the model bias was reduced further to $-3.1\,\%$. An alternative approach to model bias reduction based on tuning the model's temporal emission profiles was introduced. By redistributing the model's $NO_x$ emissions accordingly, a nighttime bias of $+27.7\,\%$ was obtained, while the noontime bias increased to $+24.1\,\%$. Optimizing the temporal profiles alone is therefore less promising than optimizing mixing, also because model-external validation of the chosen profiles can not be provided at this point. Nonetheless the study demonstrates, how sensitive modelled surface $NO_2$ concentrations are to the temporal emission profiles. This is an important finding, seeing that significant corrections to the monthly or weekly temporal profiles were derived in model evaluation papers of the past (see e.g. Poraicu et al. (2023); Kumar et al. (2021)), implying that corresponding uncertainties should be assumed for the diurnal profiles (and in consequence, the model results), as well. The $NO_2/NO_x$ ratios in the model could be improved by investigating other chemical mechanisms, photolysis schemes, and speciation profiles in sensitivity studies similar to those presented in our study.

In sect. 3.2 modelled tropospheric $NO_2$ VCDs were compared to TROPOMI measurements. In order to make a representative comparison, the air mass factors of the TROPOMI retrieval were recomputed using the high-resolution $NO_2$ profiles from our simulation. The distribution of modelled and observed $NO_2$ VCDs was found to be similar and agrees with previously reported simulation results. It was shown, that the model tends to overestimate the $NO_2$ VCDs and surface concentrations in similar geographic regions, e.g. in the strongly polluted regions of west Germany and the Netherlands. The five different model setups were shown to perform similarly well in the evaluation against satellite data, with biases between $+3.0\,\%$ and $+13.8\,\%$, RMSE values of $0.80 \cdot 10^{15}\,\mathrm{molec\ cm^{-2}}$ to $1.14 \cdot 10^{15}\,\mathrm{molec\ cm^{-2}}$, and $R$-values between 0.87 and 0.89. The model run with enhanced mixing showed reduced bias over the polluted regions of West Germany, whereas the model run with tuned temporal emission

profiles showed stronger overestimations and performed worst overall. The results of sect. 3.2 can be seen as a validation in favour of the model run with enhanced mixing. An important finding of the study is the overall weak sensitivity of the modelled $NO_2$ VCD to the choice of model parameters, because it demonstrates that good general agreement with satellite data alone is not sufficient for the validation of a WRF-Chem simulation.

In sect. 3.3 modelled noontime $NO_2$ concentration profiles were compared to profiles from MAX-DOAS measurements at five locations in central Europe from the FRM4DOAS network. The inversion algorithms MMF and MAPA were used. The level of agreement between simulations and observations varied from location to location. In most locations, the overall shape of the profile was reproduced well, but (mostly positive) biases occurred. The model run with enhanced mixing, which performed best in the remaining comparisons, showed biases of $+12.0\,\%$ (Mainz), $-25.3\,\%$ (Bremen), $-5.9\,\%$ (Heidelberg), $+50.7\,\%$ (Uccle), and $+3.9\%$ (De Bilt). Two main qualitative differences between observed and simulated profiles were identified: Firstly, the simulated profiles showed steep $NO_2$ gradients close to the surface and agreed well with the collocated surface concentration measurements. The observed profiles did not resolve the surface layer well, most likely due to the coarse retrieval grid with 200 m layer height. As a consequence, the observed profiles tended to underestimate the surface concentration measurements. Secondly, elevated $NO_2$ layers were found in some of the observed profiles, but not in the corresponding simulated profiles. Here, the lack of vertical emission profiles, the overall spatial resolution of our WRF-Chem simulations, and the influence of horizontal concentration gradients on the MAX-DOAS inversion procedure were identified as possible explanations.

For future research, a definite statement on the issue of $NO_y$ cross sensitivities of in-situ measurements should be given by the UBA. The WRF-Chem modelling community should reconsider the current use of clipping thresholds for the mixing coefficients, seeing that their implementation is not transparent to the user and it is unclear how their original values were obtained. The parametrization should at least be made accessible through the namelist interface. In addition to the modification of the vertical mixing scheme, also a combined optimization of mixing and temporal profiles seems promising and should be tested in the future. Introduction of vertical emission profiles could potentially yield better agreement with MAX-DOAS measurements, but likely requires re-adjustments of the lower mixing thresholds, which were deducted for simulations without vertically distributed emissions. Further model validations could be drawn from running the WRF-Chem simulation at the spatial resolution of the emission inventory. At least over Germany, where emission data with a resolution of 1 km $\times$ 1 km is available, the model resolution could in theory be increased by a factor of 9. It is possible that at such resolutions, the "traffic" measurement stations, which were dismissed in this study, can be used in a sensible way. However, this would realistically require to shrink the geographical extent of the model domains. The feasibility of tuning the temporal emission profiles could be re-evaluated with the help of geostationary satellites e.g. GEMS (Kim et al., 2020), Sentinel-4 (Stark et al., 2013), and TEMPO (Naeger et al., 2021). With an hourly measurement frequency, diurnal emission profiles could be derived with broad geospatial coverage and used for validation. Furthermore, the results of sect. 3.2 would be much more informative, if satellite observations were available for different hours of the day. The synergistic use of next-generation satellites and RCT models of much higher resolution has promising potential to expand the studies presented in this article.

*Data availability.* All data are available from the authors upon request.

575   *Author contributions.* LK, TW, SB, and VK developed the question of research. VK, LK, SO, and RK conducted the WRF-Chem simulations. LK made the comparison between simulations and observational datasets. LK wrote the article, with all authors contributing by revising it interactively.

*Competing interests.* The authors declare that they have no conflict of interest.

*Acknowledgements.* We acknowledge the Umweltbundesamt and Deutscher Wetterdienst for providing in-situ measurement data and the 580   UBA emission inventory. Furthermore, we acknowledge Udo Frieß and François Hendrick for providing access to and guidance through the FRM4DOAS datasets. Data analysis and visualization were performed in Python 3.9 including standard libraries such as numpy, scipy, pandas, netCDF4, matplotlib, and basemap. For processing of emission data the HERMES software was used. One of the co-authors (R. Kumar) is from the National Center for Atmospheric Research, which is sponsored by the National Science Foundation of the United States.

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

**Appendix A: Additional figures and tables**

**Table A1.** Layer extent of the lowest 24 layers in our WRF-Chem simulation

| Layer number | Layer bottom [m] | Layer top [m] | Layer number | Layer bottom [m] | Layer top [m] |
|---|---|---|---|---|---|
| 1 | 0 | 8 | 13 | 1370 | 1546 |
| 2 | 8 | 33 | 14 | 1546 | 1697 |
| 3 | 33 | 66 | 15 | 1697 | 1841 |
| 4 | 66 | 125 | 16 | 1841 | 1937 |
| 5 | 125 | 209 | 17 | 1937 | 2035 |
| 6 | 209 | 310 | 18 | 2035 | 2183 |
| 7 | 310 | 429 | 19 | 2183 | 2374 |
| 8 | 429 | 575 | 20 | 2374 | 2661 |
| 9 | 575 | 741 | 21 | 2661 | 3142 |
| 10 | 741 | 935 | 22 | 3142 | 3907 |
| 11 | 935 | 1178 | 23 | 3907 | 4762 |
| 12 | 1178 | 1370 | 24 | 4762 | 5643 |

Exact layer bottoms and tops depend on location and time. The values given here are averages.

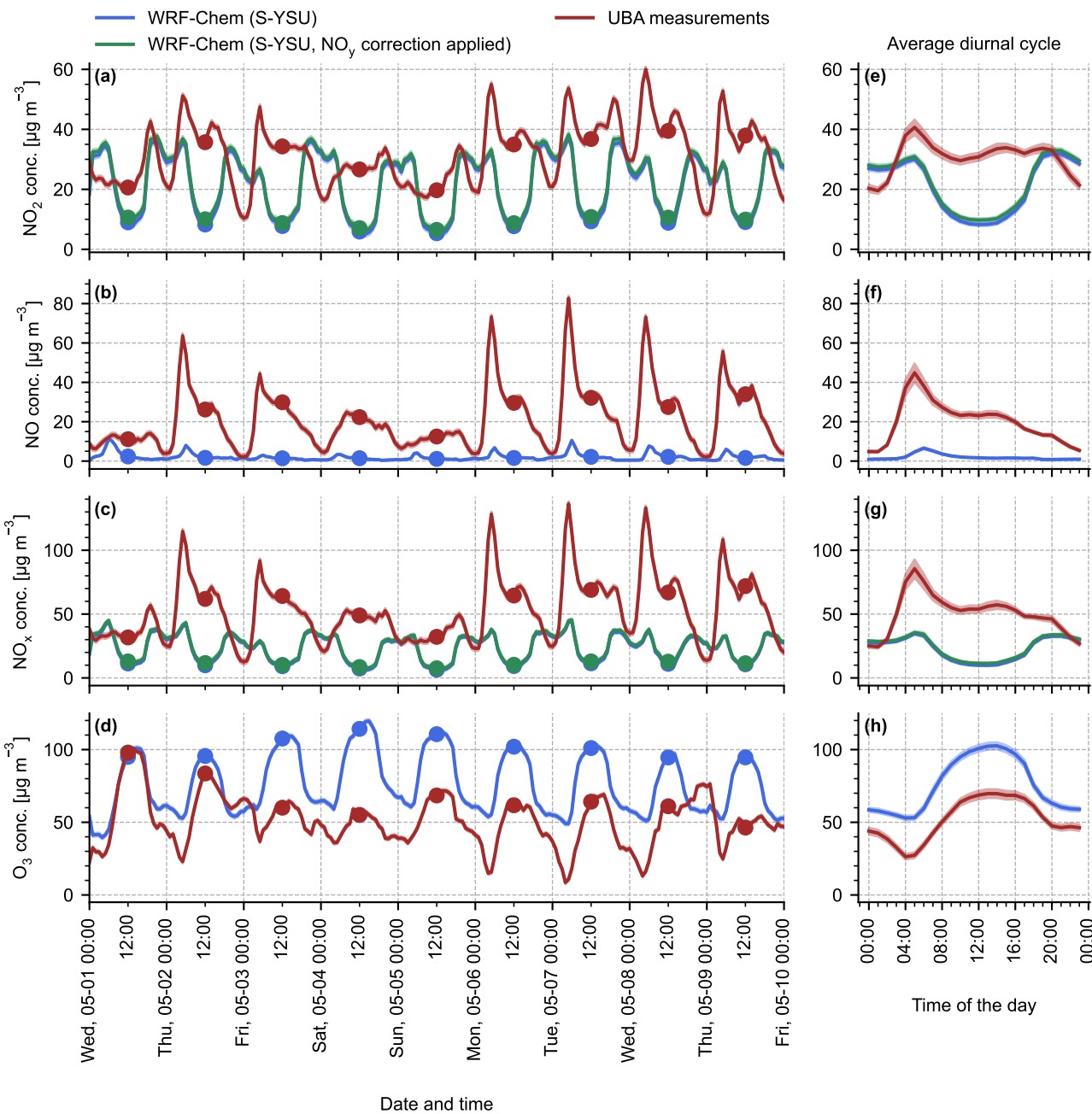

**Figure A1.** Like Fig. 3, but for traffic stations instead of background stations.

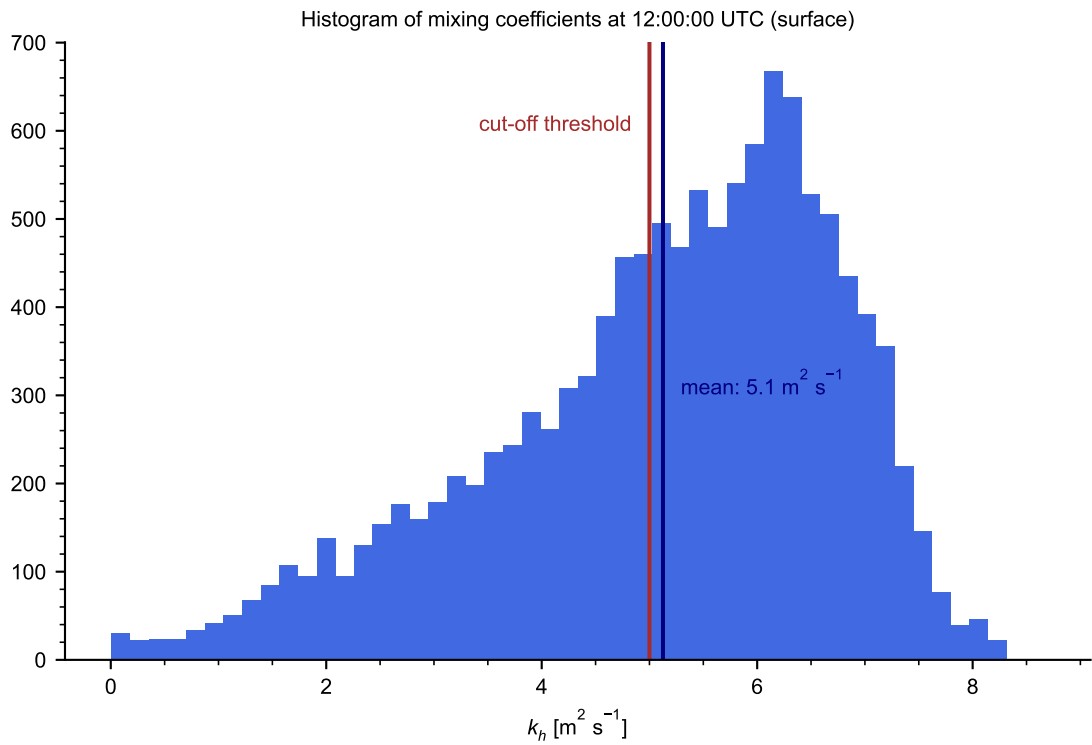

**Figure A2.** Histogram of the mixing coefficients used to produce the average values displayed in Fig. 6 for the simulation run S-YSU at noontime.

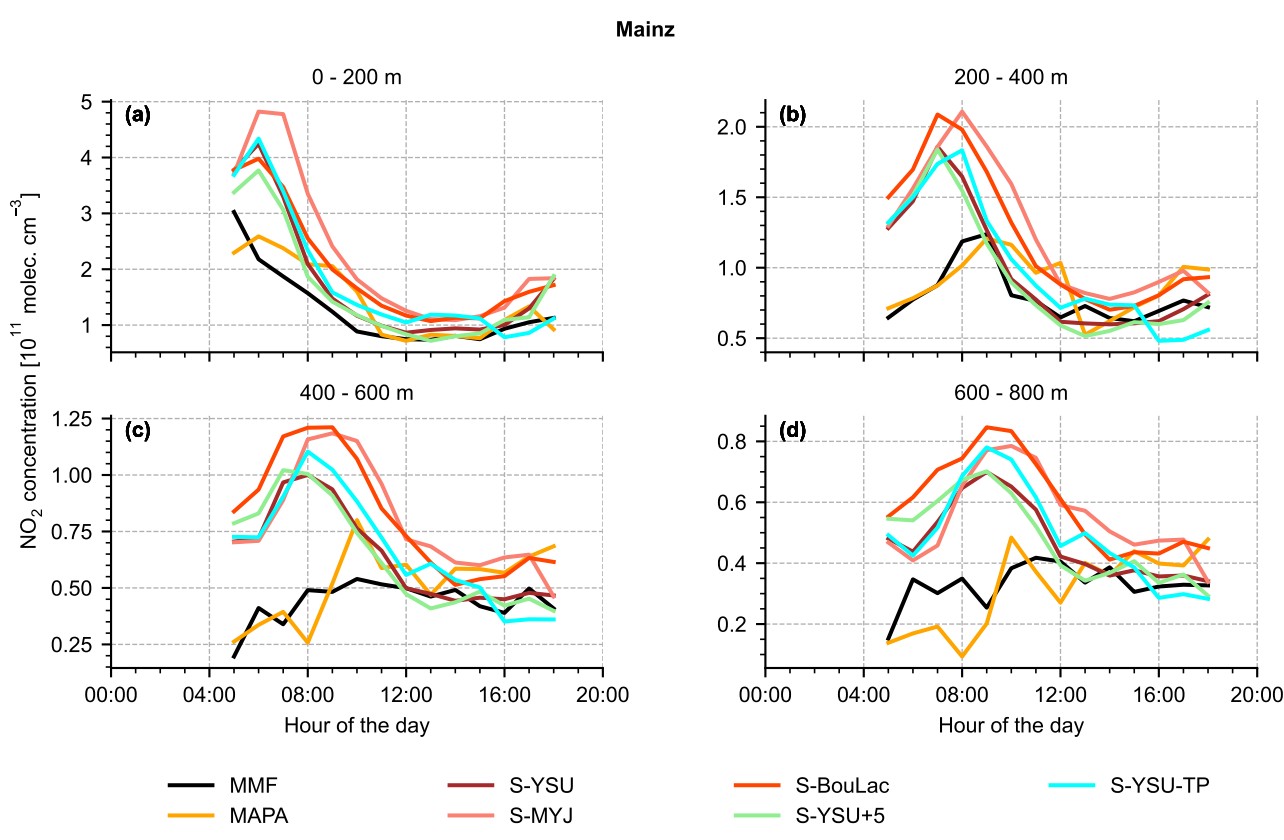

**Figure A3.** Temporally resolved comparison between $NO_2$ concentration profiles from MAX-DOAS measurements in Mainz, Germany, and simulated profiles from WRF-Chem. Here, only the first four layers of the MAX-DOAS retrieval are shown, one layer per subplot.

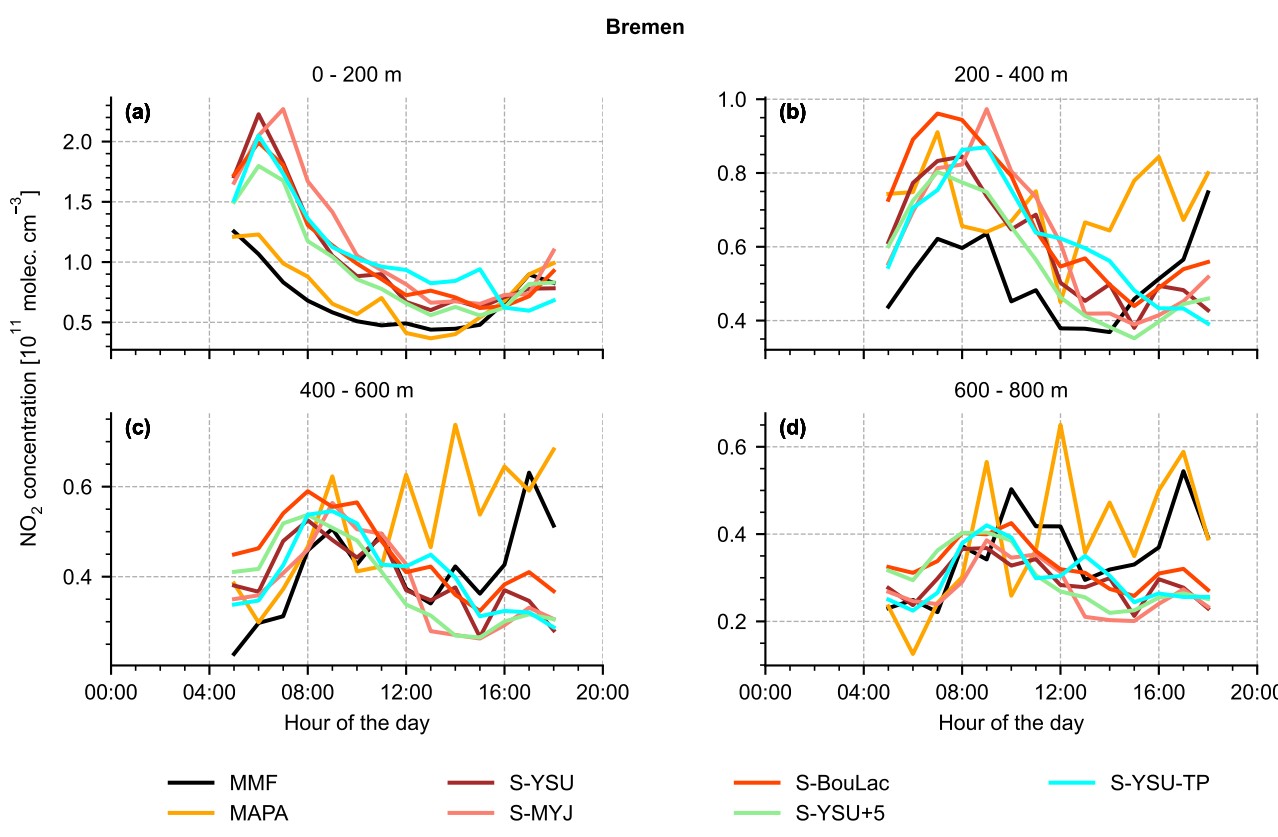

**Figure A4.** Like Fig. A3, but for Bremen, Germany.

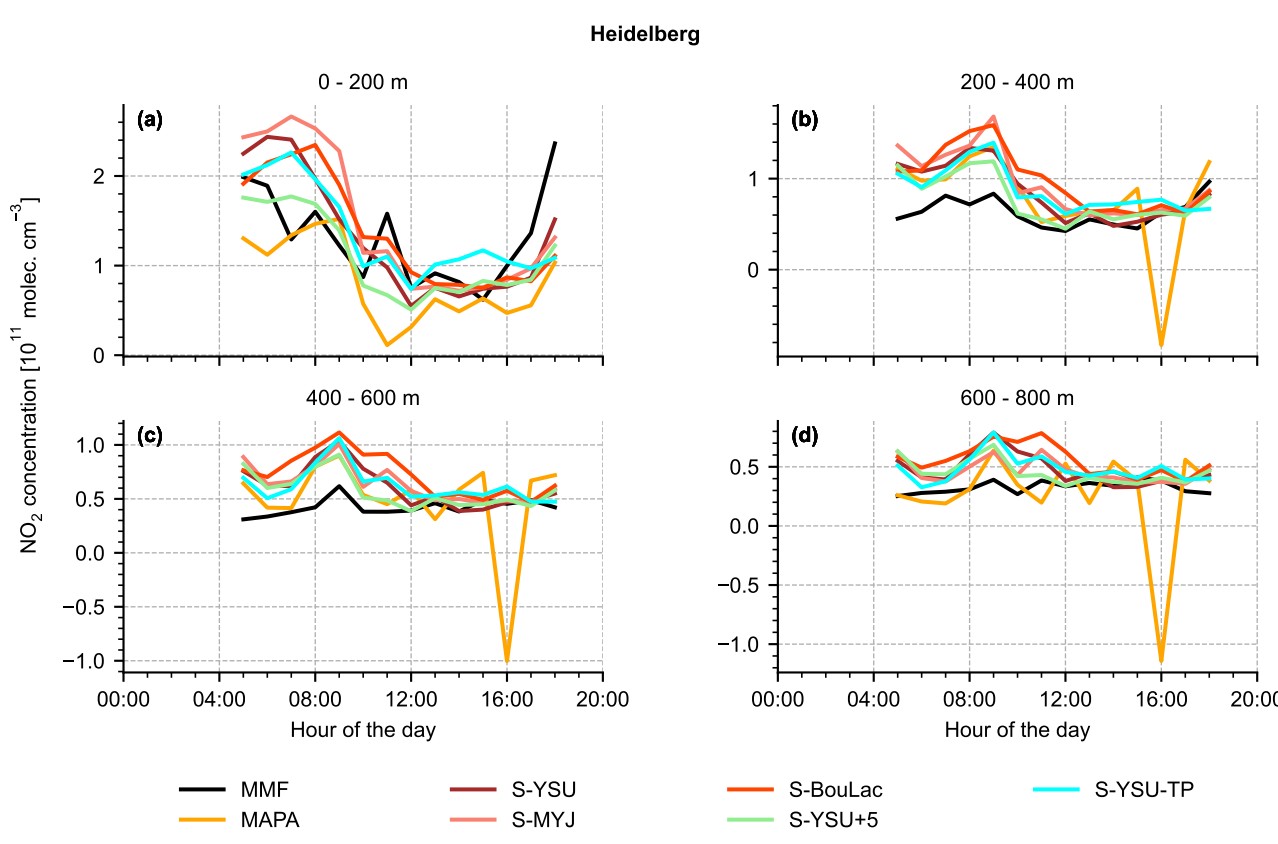

**Figure A5.** Like Fig. A3, but for Heidelberg, Germany.

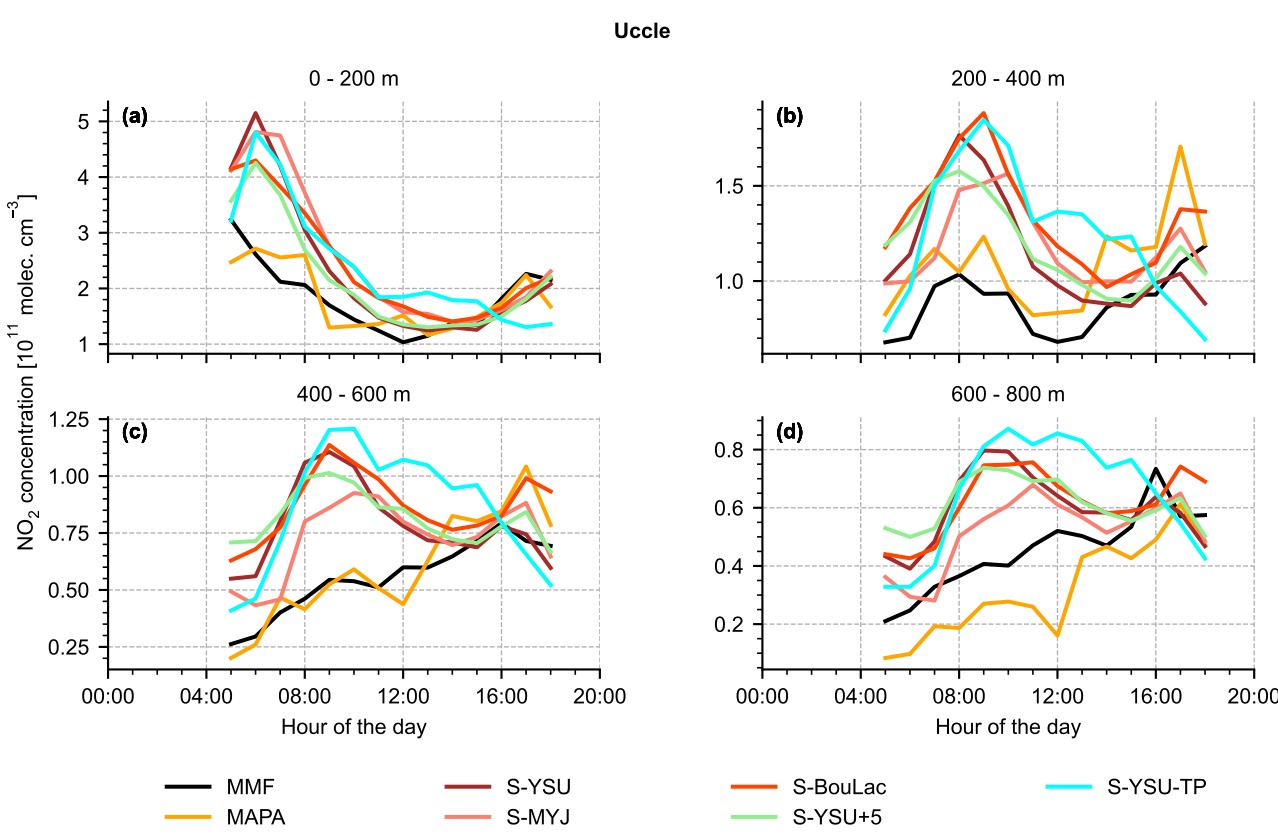

**Figure A6.** Like Fig. A3, but for Uccle, Belgium.

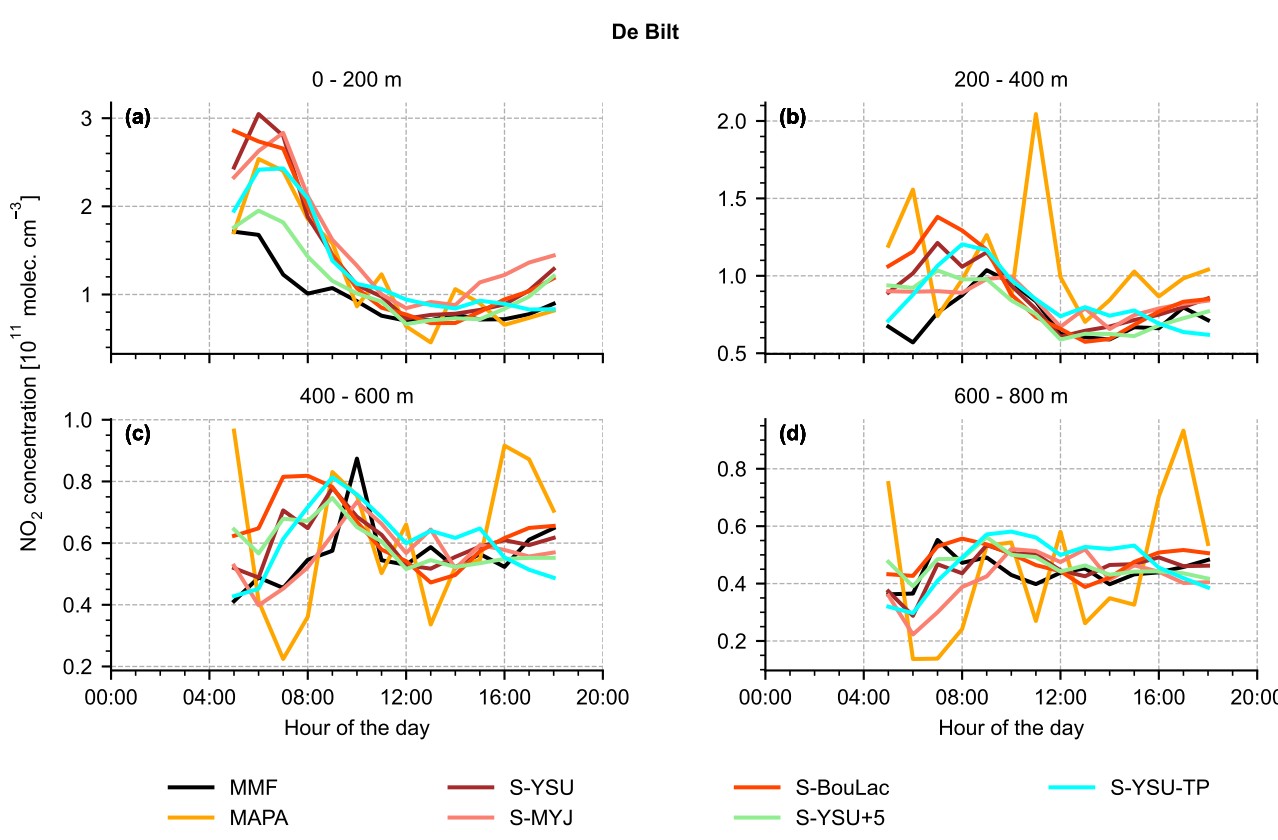

**Figure A7.** Like Fig. A3, but for De Bilt, Netherlands.