# Peer review of "On the influence of vertical mixing, boundary layer schemes, and temporal emission profiles on tropospheric $NO_2$ in WRF-Chem - Comparisons to in-situ, satellite, and MAX-DOAS observations"

_EGUsphere, 2022_

## Author Comment (AC1)

**Authors' reply to 'Reviewer comments on egusphere-2022-1473'**

The authors would like to thank the Referee #2 for the constructive review of the manuscript. The Referee has raised several points of criticism, and the authors agree to many of them. As a consequence, the manuscript has been revised from the ground up. The major changes in the revised version are:

- Additional model sensitivity studies w.r.t. the boundary layer and surface layer scheme were added.

- The $NO_y$ cross sensitivities of the in-situ reference measurements were taken into account.

- The influence of vertical mixing was investigated, and an alternative model run with optimized vertical mixing is presented. Although this model simulation still relies on parameter optimization, it can be better justified from a technical perspective. It also produces better overall results than the modelling strategy of the original manuscript. The results of this model run are considered the main finding of the revised manuscript.

- As a consequence, the temporal emission profiles are no longer the main focus of the manuscript. They remain in the revised manuscript in the form of a sensitivity study, but are no longer presented as the definite solution to resolving model biases. The title of the paper has been changed to:

   *On the influence of vertical mixing, boundary layer schemes, and temporal emission profiles on tropospheric NO₂ in WRF-Chem - Comparisons to in-situ, satellite, and MAX-DOAS observations*

The authors hope, that the revised manuscript is convincing and addresses the Referee's criticism adequately. Below, the authors respond to the review point by point. The referee's comments are printed in blue, and the authors' responses in black.

— — —

**Major comments:**

1) Several features of the adjustment seem suspicious, such as the disappearance of the late afternoon traffic peak (the 17-19h emissions are decreased by about a factor of 2!), the disappearance of the early evening "energy for building" peak, the pronounced early afternoon enhancement for the industrial and energy sectors, etc. The authors do not provide any validation (not even a discussion) for these changes. The changes are claimed to be reasonable, but this needs to be demonstrated.

The model run with tuned temporal profiles is no longer at the heart of the revised manuscript. All other runs in the revised version use the temporal profiles from Kumar et al. (2021), which are derived from observations and generally considered valid in the modelling community. The authors have decided to keep the run with tuned temporal profiles in the manuscript in the form of a sensitivity study. A study of this kind was suggested elsewhere, e.g. by Poraicu et al. (2023), who expressed that the model sensitivity to diurnal emission profiles should be investigated. Thus we found it useful to keep this model run in the paper.

2) Using another model (MECO(n)), Kumar et al. (2021) succeeded in reproducing quite well the NO2 diurnal cycle from MAX-DOAS stations (also in situ data, to a lesser extent). No adjustment of the emission temporal profile was found necessary to achieve this result

Kumar et al. (2021) do not present temporally resolved comparisons to in-situ data, except for a single station (see Fig. 13 (c), (e)). Their comparison in Fig. 4 shows monthly means. Such comparisons can not be used for an assessment of how well a model reproduces diurnal cycles of trace gases, or how well-adjusted the temporal emission profiles are.

As addressed in major comment 3 below, the mixing routine of WRF-Chem implements highly questionable manipulations of the vertical mixing coefficients. The MECO(n) model most likely treats vertical mixing much differently, hence why Kumar et al. (2021) showed convincing results without further tuning of the temporal profiles.

3) Many WRF-Chem studies have shown that the simulation of vertical mixing (especially during the night) plays an important role in contributing to the model bias (Kuik et al. 2016, 2018; Visser et al. 2019; Du et al., 2020; Poraicu et al. 2023; etc.).

The Referee rightfully hints towards the importance of vertical mixing. Because this comment has influenced the results of the revised manuscript the most, it requires a detailed reply:

In WRF-Chem, the vertical mixing coefficient $k_h$ is obtained from the boundary layer scheme (higher $k_h \rightarrow$ stronger mixing). However, the mixing routine of WRF-Chem (see `chem/dry_dep_driver.F` line `690` ff.) implements a parametrized manipulation of $k_h$ by setting it to a minimum value of 1 over rural regions and 2 over urban regions. In the revised manuscript (see Fig. 6) it is shown that particularly at nighttime, the boundary layer schemes YSU, MYJ, and Bougeault-Lacarrere produce mixing coefficients far smaller than 1 at the surface. Therefore the current implementation effectively enforces overall stronger mixing.

Du et al. (2020) discussed this issue and implemented code changes on the boundary layer module level increasing the lower mixing threshold further to a value of 5 (instead of 1, and 2, respectively). The revised manuscript shows a simulation run, following the recommendation of Du et al. (2020), although with a more generalized implementation. This run produces the overall best simulation results, and the authors consider this the main finding of the revised manuscript, particularly because, to the knowledge of the authors, the outdated mixing procedure is not discussed in any publications other than Du et al. (2020).

Related changes in the revised version:

- Section 2.2 explains additional model runs in which the influence of different combinations of boundary layer and surface layer schemes are addressed.

- Section 2.3 explains the additional model run with enforced mixing according to Du et al. (2020).

- Section 3 describes and compares the results obtained from the new model runs.

- Table 2 gives an overview of the new model runs.

- Figure 5 compares the diurnal cycle of modelled $NO_2$ surface concentrations of all model runs.

- Figure 6 displays the diurnal cycles of the boundary layer height and the mixing coefficient $k_h$ across different model runs in the context of Du et al. (2020).

- Figure 7 shows vertical profiles of $k_h$ at different hours of the day.

- Figure 10 compares the modelled $NO_2$ VCDs of all model runs.

- Tables 3, 4, 5, and 6 summarize the results of all simulation runs.

- Figures 13 and 14 now also show the results of the new simulation runs.

- Section 4 discusses the results, where the authors have underlined the importance of vertical mixing and treat the original approach of tuning temporal emission profiles more critically.

4) Studies have also shown the importance of the chemical mechanism, e.g. Knote et al. 2015. Furthermore, the VOC emissions in the model are not well validated and could have a large impact on the NOx, due to their effects on OH radical levels and organic (peroxy)nitrate formation.

The Referee is certainly correct about the possible impact of the chemical mechanism. The authors believe that intercomparisons of chemical mechanisms should be addressed in a separate paper. This is due to the general complexity of the problem; publications such as Knote et al. (2015) compare chemical mechanisms in a highly simplified model environment (essentially a large box model) for that reason.

An attempt to compare modelled and satellite-observed HCHO columns for implicit validation of VOCs is found in the reply to minor comments below.

Related changes in the revised version:

• Section 1 describes the impact of the chemical mechanism more adequately.

5) Most importantly, the NO2 measurements used at 92% of the German sites are made using chemiluminescence instruments, which are characterized by sometimes large positive biases due to interferences of other NOy compounds such as PAN, HNO3 and HONO (Lamsal et al., 2008; Villena et al., 2012). As shown by Villena et al. (2012), the bias can reach up to a factor 4 and correlates with ozone, presumably because ozone correlates with photochemical activity. As discussed above, the emission update proposed by Kuhn et al. consists essentially in a significant increase during daytime, especially between 8 and 17h (Figure 2), precisely when ozone is most abundant (Figure 3). It is very unlikely that this would be a coincidence.

Referee 2 rightfully points out the issue of cross sensitivities to other nitrogen species (such as PAN, HONO$_2$, and alkyl nitrates) in molybdenum converters.

In the revised manuscript, the correction factor of Lamsal et al. (2008)

$$F = 1 + \frac{0.35 \cdot [\text{PAN}] + 0.95 \cdot [\text{HNO}_3]}{[\text{NO}_2]}$$

is computed, where [PAN], [HNO$_3$], and [NO$_2$] are the volume mixing ratios of PAN, HNO$_3$, and NO$_2$, respectively. Note, that the minor contribution of alkyl nitrates can not be included, because they are not available in the MOZART-4 chemical mechanism. The modelled NO$_2$ concentrations are multiplied with $F$ before comparison to the reference measurements. Alternatively, the reference measurements could be divided by $F$, but this would result in slightly different reference values for each model run. The correction is applied exclusively in the comparison to molybdenum-based instruments. This method is identical to that of Poraicu et al. (2023).

Related changes in the revised version:

• Section 1 describes the issue of NO$_y$ cross sensitivities more critically.

• Section 3.1 describes the implementation of the correction factor $F$.

• Figure 3 (a), (b) show the diurnal cycles of NO$_2$ and NO$_x$ with and without the application of $F$.

• Figure 4 shows the diurnal cycle of $F$.

• Figure 5 (a), (b) show the diurnal cycle of NO$_2$ of all simulation runs with and without the application of $F$.

Furthermore, the claimed model improvement against TROPOMI is far from being evident, judging from Figures 5g and 6h (also the slope and RMSE, see Table 4). The NO2 model overestimations over the Rhine/Ruhr and other areas are lower with the original temporal profile from Kumar et al. (2021).

The overall weak sensitivity of the modelled NO$_2$ VCD to the choice of model parameters is an important finding of the study: It demonstrates that good general agreement with satellite data alone is not sufficient for the validation of a WRF-Chem simulation. This is shown by comparing

Fig. 10 and Fig. 5 (b). While the satellite comparisons in Fig. 10 show mostly similar results, the surface concentrations in Fig. 5 (b) are entirely different.

Comparison of the different simulation runs (see Table 5) shows, that the new simulation run with enforced mixing shows the lowest RMSE and decreased overestimations in west Europe, complementing the results of section 3.1. The authors argue, that in this case, the model improvement is evident. The manuscript no longer claims superiority of the tuned temporal emission profiles w.r.t satellite data.

Finally, the model with updated profiles overestimates the MAX-DOAS $NO_2$ data at most sites, and by as much as 30, 50 and 79% at Heidelberg, Mainz and Uccle. Evaluation of the diurnal cycle of modelled $NO_2$ against the MAX-DOAS data is missing, as well as the assessment of the effect of the new profiles on the model agreement against MAX-DOAS data.

It should be noted that the difference between the two reference datasets MMF and MAPA are in many cases just as large. In that sense, disagreements of 30-50 % are not exceptional. The deviations and their possible explanations were also discussed at length in the original manuscript.

Related changes in the revised version:

• Figures 13 and 14, and Table 6 now show the results of all simulation runs

• The temporal interpolation required for profile comparison was improved (from nearest neighbor to linear), which yields slightly different results (e.g. mean bias at Heidelberg changed from +30.1 % to +27.2 %)

• Appendix Fig. 2-6 show a comparison of the diurnal cycle of modelled $NO_2$ against MAX-DOAS data in the lower layer of the retrieval algorithms, as requested by the Referee.

In conclusion, I cannot recommend publication of the manuscript in its present form. The WRF-Chem comparisons with in situ, MAX-DOAS and TROPOMI data are interesting, but the emission adjustment (which is the main point of the paper) has no added value. I suggest to show results with the original diurnal profiles of Kumar et al., and to explore the various possible causes for the biases, possibly through additional sensitivity calculations.

The authors have followed the Referee's suggestion in the revised version, as explained thoroughly above and below.

**Minor comments:**

l. 8 "crucial aspects of the retrieval etc.": the application of averaging kernels has become quite standard in model evaluations against UV-Vi satellite data. As I understand, you only follow the standard recommendations. Please re-phrase.

The sentence was changed to:

*A comparison between modelled $NO_2$ vertical column densities (VCDs) and satellite observations from TROPOMI (TROPOspheric Monitoring Instrument) is conducted, with averaging kernels taken into account.*

l. 46 Mar et al. (2016): "the choice of mechanism barely impacts the NOx underestimation": this statement is based on a comparison of only 2 mechanisms! This cannot begin to describe the uncertainties related to the chemical mechanism. For example, both mechanisms considered might have similar flaws, such as the absence of HONO chemistry (and heterogeneous production), outdated organic nitrate chemistry, overestimated rate constant for NO2+OH (see e.g. Mollner, 2010), etc. Please re-phrase or drop the sentence.

The section was rephrased, and now states the significant deviations between chemical mechanisms found by Knote et al. (2015):

*Mar et al. (2016) study the influence of the chemical mechanism on modelled $O_3$ and $NO_2$ by direct comparison of the mechanisms MOZART (Model for OZone and Related chemical Tracers) and RADM2. While the two mechanisms were found to produce significantly different results for $O_3$, the differences in modelled $NO_2$ were much smaller. On average, the $NO_2$ concentrations obtained*

*from MOZART were 2 µg m$^{-3}$ larger than those obtained from RADM2. However, a study based on box-model simulations by Knote et al. (2015) reveals much larger discrepancies between chemical mechanisms in WRF-Chem of up to 25 % for $NO_x$ and 100 % for $NO_3$, which plays a significant role for nighttime $NO_x$ chemistry. Furthermore, specific parts of chemical mechanisms in WRF-Chem were found to be outdated, such organic nitrate chemistry, or the rate constant of OH and $NO_2$ to form gaseous nitric acid ($HONO_2$, see Mollner et al., 2010).*

l. 61 "a systematic investigation to prove a general bias of the NO2 in-situ measurements in Europe is still missing": however, Villena et al (2012) showed an overestimation of up to a  factor 4 compared to DOAS NO2 measurements. The overestimation peaks in the afternoon when ozone is at its highest levels, which is also when the WRF-Chem model NO2 underestimation is highest compared to the network measurements is highest.

An overestimation of +400 % is most likely an outlier, as the other publications (cited in the revised manuscript) have found far more moderate biases in the range of +20 % to +50 %. The revised manuscript describes the problem of cross-sensitivities in the in-situ measurements in much larger detail than originally:

*A number of publications show that the observed daytime low bias of modelled surface $NO_2$ could relate to systematic flaws in the groundbased in-situ measurements used as reference. Conventional in-situ methods often utilize molybdenum converters, which were found to be cross sensitive to other reactive nitrogen species, such as PAN, $HNO_3$, and alkyl nitrates, summarized as $NO_y$. This issue was discussed e.g. by Dunlea et al. (2007), Steinbacher et al. (2007), Lamsal et al. (2008), Boersma et al. (2009), and Villena et al. (2012), who found biases reaching up to a factor of 4 with a strong correlation to $O_3$ (which again correlates with photochemical activity). Poraicu et al. (2023) attempt to account for such cross sensitivities by computing a correction factor based on simulated surface mixing ratios of PAN and $HNO_3$ and the empirical estimates of the relevant conversion efficiencies as reported by Lamsal et al. (2008). The method contributes to resolving the daytime low bias of the model, but is not helpful with respect to the even larger high bias at nighttime. In Europe, in-situ measurements of $NO_2$ must conform to regulations defined by the European Norms 14221, 14181, and 15267, which require empirical evidence that the instrument in question is unbiased against direct (e.g. spectroscopic) measurements of $NO_2$. Such conformity assessments are conducted and thoroughly protocolled by technical inspection associations (such as the TÜV for the in-situ measurements in Germany, which are used in this article, see German Environmental Agency (a)). There is a clear conflict between the overestimations reported in the scientific literature and the proclaimed conformance to the European regulations and the true magnitude of the problem remains up for question.*

l. 62 It is not because the instruments "follow strict regulations" that they are free of artefacts. The technique itself leads to the presence of interference.

See above.

The overestimation of 7% in test measurements mentioned in the manuscript does not validate the technique, because the interference is highly variable, as shown by Villena et al. (2012). A "personal communication" is impossible to verify and place in proper context. The variability in the bias due to interferences appears related to photochemical activity.

The assessment in question was issued by the German federal agency, which must conform to the European norms listed above. As such, these statements do have weight and can not be disregarded so easily. The reference in question was changed to the official documentations of the TÜV (only available in German language), on which the assessment of the UBA was based.

l. 76-77 The "validation" of the temporal profiles of emissions using TROPOMI and MAX-DOAS is of very limited value since TROPOMI measures only in early afternoon, and only midday MAX-DOAS data are used in this manuscript.

As described, the temporal profiles are secondary in the revised manuscript. The diurnally resolved comparison of modelled $NO_2$ profiles of MAX-DOAS data was added in Appendix Fig. 2-6.

l. 204 scaling of the O3 BCs : a uniform scaling? Is this based on ozonesonde data or only surface data? How about the vertical dimension? Why not use CAMS profiles ? Those are likely more realistic than CAM-chem.

This section was removed, seeing that it yields no real improvement to the simulated $NO_2$ and leads away from the main statements of the manuscript.

l. 224 Why would the uncertainty on simulated surface concentration be equal to the maximum hour-to-hour variation in concentration? This does not make sense.

The uncertainty of the simulated surface concentration is now computed as what is conventionally called the „uncertainty of the mean".

l. 245-246 'the temporal profiles (...) for the traffic sector (Fig 2a) have a shape similar to the NOx time series of the traffic stations (...) This can be seen as further validation of our results." : certainly not, since the NOx concentrations temporal profile reflects not only the emission profile but also the temporal cycle of boundary layer mixing.

The authors agree, and the statement has been removed.

The section 3.1 could be rewritten by making more rapidly your point that the new simulation performs better, since the optimization of profiles was designed to do that, and it is not surprising at all that the results reflect that. On the other hand, you should explain why your new emission profiles can be considered more realistic.

Section 3.1 covers far more aspects in the revised manuscript. The authors have put the focus on demonstrating the benefits of the enforced mixing approach and intercomparing the different new simulation runs.

L. 285 The increase in NO2 above 10 km altitude is not necessarily a result of stratosphere-troposphere exchange. Higher NO2 in the UT (compared to the mid-troposphere) could result from higher NOx lifetime, aircraft emissions or lightning emissions. I recommend to use the best estimate of the tropopause level. What is the impact of removing 3 layers from the calculated tropospheric column?

The authors follow the recommendation and use the full tropospheric profile instead. Overall, the results slightly improve (higher $R$, lower RMSE, lower relative biases)

Related changes in the revised version:

• Eq. (7), (8) were changed so the summation runs up to the tropopause layer index.

• Table 5 contains the updated results of the satellite comparison.

• Appendix Fig. 2 (original manuscript) has been removed, as it no longer serves any purpose

- The satellite comparison was changed to use the newest product version (20400) instead of the PAL dataset.

Based on the updated satellite comparison it can be estimated, that the uppermost 3 layers of the troposphere contribute by approx. 10 % to 15 % to the total simulated column.

l. 300-308 This discussion could be shortened, since you are simply applying the standard recommendation issued for TROPOMI NO2.
The panels (k)-(o) of Figure 6 would be better placed beneath the panels (f)-(j) of Figure 5. In this way, the reader can better judge the impact of the new temporal profiles on the comparisons with TROPOMI, which is the purpose of this section. The improvement of results due to the application of averaging kernels is a pretty standard result.

The authors find the explanation brief enough to keep it in the manuscript. Readers who are not familiar with the processing of satellite data may find this helpful.

The comparison to satellite data was restructured. In the revised manuscript, the impact of re-computing the air mass factors is shown in Fig. 9, and the intercomparison of model runs in Fig. 10, stacked vertically as requested by the Referee.

l. 331 Based on the figures 6(m) and (5)h, the simulation using the Kumar profiles performs better. Is the low bias of -15.7% the mean relative bias, or the relative bias of the mean column? The results shown in Table 4 suggest a better slope and RMSE with the old profiles. The slightly larger bias could be due to an emission underestimation.

The authors agree, that the model run with tuned temporal profiles did not perform better in the comparison to satellite data. The RMSE values across the runs with temporal profiles from Kumar et al. (2021) are in the range of $(0.80 - 0.93) \cdot 10^{15}$ molec. cm$^{-2}$, but significantly higher $(1.14 \cdot 10^{15}$ molec. cm$^{-2})$ in the run with tuned temporal profiles. The simulation results in the revised manuscript show overall more convincing differences to the base run. In particular, the run with enforced vertical mixing performs best, with the lowest RMSE of $0.80 \cdot 10^{15}$ molec. cm$^{-2}$ (vs. $0.85 \cdot 10^{15}$ molec. cm$^{-2}$ in the base run). „Bias" refers to the mean relative bias, as defined in eq. (4). The authors argue, that RMSE should be the metric of choice (as opposed to e.g. slope).

l. 339 "In comparison to monthly means, the modelled NO2 VCDs are smeared out": is this what you really mean? Please re-phrase.
The paragraph 336-345 brings very little to the discussion. The stronger noise is expected. Consider removing that part (and the figure)

The section has been removed, as requested by the Referee.

l. 389 "Elevated layers are typically caused by elevated emissions, e.g. from a power plant stack at a few hundred meters height": Is there really a tall power plant stack in the vicinity of every MAX-DOAS station ?

This is not the case, and a conflicting sentence „This occurs regularly at all MAX-DOAS stations in this study (…)" was removed. A clear case of elevated emissions is found in Bremen, for where Bösch (2018) discussed the influence of a local power plant. The NO$_2$ profiles obtained with MMF could also hint towards elevated emissions in Heidelberg, where a local power plant exists in viewing direction of the instrument. Uccle shows elevated concentrations for MAPA, but not for MMF.

l. 446 Overestimation of NO2/NO: might reflect the fact that interferences affect only NO2, not NO.

The revised manuscript, where interferences were taken into account, still shows an overestimation of NO$_2$/NO (see Fig. 3 (e), (f)). The aspect of NO$_2$/NO ratios was brought up and discussed in the authors' reply to 'Comment on egusphere-2022-1473' by Referee #3.

I. 450 Evaluation of the VOC emissions could be done using TROPOMI HCHO columns.

The evaluation of VOC emissions using TROPOMI HCHO columns is hindered by the noisiness of the satellite observation on our domain. The figure below shows a direct comparison of the monthly mean (May 2019) HCHO column from TROPOMI and our simulation (simulation run S-YSU+5, see revised manuscript) with qa $\geq$ 0.75.

[Figure]

Judging from the mean HCHO VCDs on the domain:
- TROPOMI: $3.95 \cdot 10^{15}$ molec. cm$^{-2}$
- WRF-Chem: $5.13 \cdot 10^{15}$ molec. cm$^{-2}$

The model overestimates HCHO VCDs by approx. 30 %. The authors would like to refer to their reply to the 'Comment on egusphere-2022-1473' by Referee #3, who have suggested a simulation run with varied VOC emissions.

I. 453-465 This discussion ignores the fact that TROPOMI NO2 data also present biases, as shown in numerous validation studies (e.g. Judd et al., 2020; van Geffen et al., 2022). Unfortunately, validation is still lacking for background conditions, but in any case, we can expect deviations of TROPOMI wrt the truth.

The biases of TROPOMI $NO_2$ data are mostly associated with the input to the retrieval algorithm, e.g. cloud information, surface albedo and $NO_2$ a priori profiles. Application of the (correct) averaging kernels together with correct $NO_2$ a priori profiles should eliminate the differences between simulated and observed VCDs. Examples for TROPOMI biases reported in literature are:

- Judd et al. (2020): -19 % to -33 % in a cloud-free case using product version v. 1.2 for late summer and autumn.
- van Geffen et al. (2022): -32 % to -23 % using product version v 2.2. These numbers include wintertime, when the biases can be expected to be more severe.

Table 5 of the revised manuscript shows a change in bias from +21.1 % to +6.7% (i.e. 14.4 % in total) between modelled and observed $NO_2$ by introduction of higher resolved $NO_2$ a priori profiles. Note, that these numbers refer to the updated satellite comparison with full tropospheric column and the newest product version (20400, see above). The authors therefore argue that the bias as described by the Referee plays no significant role in our study.

l. 468-469 "General agreement in the overall shape and magnitude": not really! There is a clear overestimation at most sites, suggesting overestimation of midday emissions. It would be enlightening to check if there is a diurnal variation of the difference between WRF-Chem and MAX-DOAS columns (or near-surface concentrations).

The overestimations occur with and without tuned temporal profiles. Therefore, they can not be explained on the basis of the tuned temporal profiles (which could lead to an overestimation of midday emissions) alone. However, Appendix Fig. 2-6 do reflect the changes introduced by tuning the temporal profiles (compare S-YSU-TP to S-YSU, e.g. in Uccle in Fig. A5 (b), (c), (d)). What is more, the influence of different boundary layer schemes seem to be of similar importance.

The authors believe that the MAX-DOAS data should not be used to evaluate emission strength. After all there are only 5 locations available, which in no way reflect the overall quality of the emission data. The comparison to the other observational data (which cover the simulation domain far better) show no signs of significant emission bias.

As requested by the Referee, the diurnal cycles of WRF-Chem and MAX-DOAS $NO_2$ concentrations for the lowest few layers of the model were plotted in Appendix Fig. 2-6.

l. 478-479 "the model accuracy (...) in RCT simulations can be strongly improved by optimizing the (...) emission profiles": sure, but probably not for the good reasons. If biases are due to other causes (measurement biases, mixing issues etc.), the updated emission profiles are worthless.

See the combined answers above. The conclusion of the manuscript (sect. 4) was rephrased to put more emphasis on the enforced mixing approach, compensation of $NO_y$ cross-sensitivities and the drawbacks of tuned temporal profiles.

---

## Author Comment (AC2)

**Authors' reply to 'Comment on egusphere-2022-1473'**

The authors would like to thank the Referee #3 for the constructive and encouraging review of the manuscript.

It must be stressed, that in 'Reviewer comments on egusphere-2022-1473' by Referee #2 several major revisions were requested, upon which the paper has been fundamentally restructured, including additional sensitivity studies to various simulation parameters. The content described by Referee #3 (mainly, the tuning of temporal $NO_x$ emission profiles) remains in the revised manuscript, but is no longer considered the main finding.

Below, the authors respond to the review point by point. The referee's comments are printed in blue, and the authors' responses in black.

———

While most of the NOx emission is set to be in form of NO, why the model simulated NO does not show significant changes and the model still misses its peaks systematically?

The simulation with tuned temporal profiles does show differences in modelled NO (see Fig. 3 (f) in the original manuscript): NO concentrations are approximately 50 % higher at daytime and 50 % lower at nighttime compared to the simulation with temporal emission profiles from Kumar et al. (2021). This agrees well with the temporal redistribution of the emissions.

The morning peak at ~ 05:30 shows almost no response to the change in emission profiles, but coincides with sunrise (see e.g. Fig. 5 in the revised manuscript). It can be expected, that photolysis of nighttime $NO_x$ reservoir species (e.g. $NO_3$ and $N_2O_5$) contributes strongly to this morning peak, hence optimization of emission profiles can not fully account for it.

The Referee has further pointed out, that the model could be overestimating the conversion rate of NO to $NO_2$, which the authors address below.

The overestimation of NO2/NO ratio suggests (authors have also pointed this out) that model is having a limitation in capturing the chemistry accurately. This needs investigation why the NO is getting converted to NO2 more rapidly than reflected in the observational data. Sensitivity simulations should be conducted with say +- 10% VOC emissions to gain insight into the likely causes and probably to reach better estimation of the diurnal variability in NOx emissions over this region.

Following the Referee's suggestion, the model run „S-YSU+5" (explained in the revised manuscript) is repeated with 10 % reduced VOC emissions. This can also be motivated by the overestimation of the HCHO column (see the authors' reply to 'Reviewer comments on egusphere-2022-1473' by Referee #2), which could indicate an overestimation of VOC emissions.

The figure below shows the diurnal cycles of modelled surface $NO_2$, NO, $NO_x$, as well as the $NO_2/NO_x$ ratio. The reduction of VOC emissions has minimal to no influence on the model results. In the original manuscript, it was shown that the reduction of $O_3$ boundary conditions by 15% results in similarly minimal changes. In consequence, it can be assumed that the oxygenation of NO to $NO_2$ (either via $O_3$ or VOCs) is not the reason for the faulty $NO_2/NO$ ratios in the model. Instead, the problem could be caused by inaccurate representations of night- or daytime chemistry in the chemical mechanism, the photolysis scheme, or the emission speciation. These points can be investigated in the future.

In the revised manuscript, the topic of $NO_2/NO$ ratios is discussed and possible solutions are mentioned.

UBA ———  S-YSU+5 ———  S-YSU+5 (90% VOC emissions) ------

---

## Author Response (AR2)

**Authors' reply to ‚Report #2 by Referee #2‘**

The authors would like to thank the Referee #2 for the constructive and encouraging report.

Below, the authors respond to the report point by point. The Referee's comments are printed in blue, and the authors' responses in black.

**Major comments:**

The clipping of the k_h would have also an effect during the day, near the top of the boundary layer, and it is plausible that k_h during daytime shows important variability and might be often below the thresold of 5 m2/s. I wonder if this could explain the significantly lower NO2 daytime concentrations of the run S-YSU+5, compared to S-YSU (Fig. 5)? This should be elucidated. My impression from the results shown is that overall, that "clipping" might be better than "no-clipping", but that the 5 m2/s threshold is a bit arbitrary and might be too high.

The Referee rightfully points out, that the mixing coefficient may fluctuate across the model domain and time, and may oftentimes fall below the clipping threshold.

At the surface, this is certainly the case, as demonstrated by the histogram below. The histogram was drawn for S-YSU at noontime at all locations of the UBA stations, i.e. the model cells from which the diurnal cycles in Fig. 5 and Fig. 6 were computed.

[Figure]

Clearly, the Referee's suspicion is correct: Enhancing the clipping threshold to $5~\mathrm{m^2~s^{-1}}$ does affect daytime mixing (although much less than at nighttime) and this is certainly the cause of the differences observed between S-YSU+5 and S-YSU during daytime.

The authors agree that the clipping threshold could be deemed arbitrary. Here it was chosen to result in a reasonable trade-off between nighttime overestimation and noontime underestimation.

The Referee rightfully points out, that the clipping should affect the top of the boundary layer, where the average mixing coefficients are close to zero at all times of the day. This may result in slightly different model behavior in the 1500 - 2000 m altitude range. However, $NO_2$ concentrations are comparably low at such altitudes and a slight manipulation of the mixing strength in these layers is expected to be negligible in our validation study.

Sect. 3.1 was changed to mention the spatiotemporal variability of the mixing coefficient and the possible effects on daytime $NO_2$ concentrations at the surface.

The histogram was added to the Appendix of the article.

My impression from the results shown is that overall, that "clipping" might be better than "no-clipping"

The authors would like to clarify that there is no case of „no-clipping" presented in their manuscript. The original WRF-Chem code already has clipping implemented. The simulation run S-YSU+5 only merely a higher clipping threshold.

**Minor comments:**

What does the clipping at 5 m2/s to the modelled ozone?

[Figure]

The left-side Figure shows the diurnal $O_3$ cycle at the surface for all relevant model runs.

Clipping the mixing coefficient to 5 m$^2$ s$^{-1}$ **increases** the nighttime $O_3$ concentration (contrary to $NO_2$, which was found to be reduced instead).

This could relate to reduced titration via reactions such as $NO_2 + O_3 \rightarrow NO_3 + O_2$, which act as an $O_3$ sink at nighttime.

Notice that the same behavior is observed equally for all other model runs, for which the clipping threshold remained at its original value.

l. 374-375 "Based on the large uncertainties of mixing and chemistry during nighttime, it is more adequate to evaluate the model perormance during daytime": I had the impression that the discrepancies at nighttime were the main motivation for the S-YSU+5, so this statement is puzzling.

This statement refers to the fact, that the $NO_x$ concentrations depend on NO, and the NO chemistry at nighttime seems to be somewhat misrepresented by the model (in particular, the build-up of the morning peak).

The sentence was removed from the manuscript, seeing that a full comparison (at nighttime / daytime / noontime) is presented directly afterwards anyways.

l. 462 Saying that total NO2 does not change at all is of course an exaggeration. Changing the vertical profile in the PBL has an impact on horizontal transport, deposition, and the convolution with the TROPOMI vertical profile of sensitivity. Please rephrase.

The sentence was rephrased to: „This is expected, seeing that in first order, the tropospheric column is a measure of total $NO_2$, which may vary only slightly across these simulation runs."

l. 443 : No, the study of Poracu et al. did not derive corrections to the diurnal cycle of emissions.

Neither the attached file with marked-up differences, nor the revised version refer to Poraicu et al. in l. 443. The authors assume that the Referee is referring to l. 525 ff. (in the marked-up differences) instead. The authors have referred to what Poraicu et al. (2022) describe in their abstract:

„Using a mass balance approach, we determined a **new weekly profile of $NO_x$ emissions** (…)"

It is therefore possible, that similar corrections will be derived for the diurnal temporal profiles in the future, which is the statement made in l. 525 ff.

l. 610: The justification for an overestimation is not convincing. Indeed, MAX-DOAS stations are located in cities, but the problem might be more an issue of representativeness than of biased emissions.

The authors have decided to remove the statement in question, seeing that a consistent overestimation of the MAX-DOAS data seems no longer evident. Except for the station „Uccle", it can be argued that S-YSU+5 represents the MAX-DOAS data altogether without no significant positive bias.

**All other minor comments and technical comments were taken into the revised manuscript as suggested by the Referee, without explicit mentioning.**